# Awareness of alcohol marketing, ownership of alcohol branded merchandise, and the association with alcohol consumption, higher-risk drinking, and drinking susceptibility in adolescents and young adults: a cross-sectional survey in the UK

Nathan Critchlow,[1] Anne Marie MacKintosh,[1] Christopher Thomas,[2] Lucie Hooper,[2] Jyotsna Vohra[2]

[1]Institute for Social Marketing, Faculty of Health Sciences and Sport, University of Stirling, Stirling, UK
[2]Cancer Policy Research Centre (CPRC), Cancer Research UK, London, London, UK

**Correspondence to**
Dr Nathan Critchlow;
nathan.critchlow@stir.ac.uk

## ABSTRACT

**Objectives** To explore awareness of alcohol marketing and ownership of alcohol branded merchandise in adolescents and young adults in the UK, what factors are associated with awareness and ownership, and what association awareness and ownership have with alcohol consumption, higher-risk drinking and susceptibility.

**Design** Online cross-sectional survey conducted during April–May 2017.

**Setting** The UK.

**Participants** Adolescents and young adults aged 11–19 years in the UK (n=3399).

**Main outcome measures** Alcohol Use Disorders Identification Test–Consumption (AUDIT-C) (0–12) and indication of higher-risk consumption (≥5 AUDIT-C) in current drinkers. Susceptibility to drink (yes/no) in never drinkers.

**Results** Eighty-two per cent of respondents were aware of at least one form of alcohol marketing in the past month and 17% owned branded merchandise. $\chi^2$ tests found that awareness of marketing and ownership of branded merchandise varied within drinking variables. For example, higher awareness of alcohol marketing was associated with being a current drinker ($\chi^2$=114.04, p<0.001), higher-risk drinking ($\chi^2$=85.84, p<0.001), and perceived parental ($\chi^2$=63.06, p<0.001) and peer approval of consumption ($\chi^2$=73.08, p<0.001). Among current drinkers, multivariate regressions (controlling for demographics and covariates) found that marketing awareness and owning branded merchandise was positively associated with AUDIT-C score and higher-risk consumption. For example, current drinkers reporting medium marketing awareness were twice as likely to be higher-risk drinkers as those reporting low awareness (adjusted OR (AOR)=2.18, 95% CI 1.39 to 3.42, p<0.001). Among never drinkers, respondents who owned branded merchandise were twice as likely to be susceptible to drinking as those who did not (AOR=1.98, 95% CI 1.20 to 3.24, p<0.01).

## Strengths and limitations of this study

► This is the first study to examine awareness of alcohol marketing and ownership of alcohol branded merchandise in a demographically representative sample of young people across the UK, including those above and below the legal purchasing age for alcohol.

► The study provides timely insight into what forms of alcohol marketing young people are aware of, how frequently they recall seeing alcohol marketing, and what factors are associated with higher awareness of alcohol marketing and ownership of alcohol branded merchandise.

► The large sample size supports robust statistical analysis to examine the relationship (if any) between alcohol marketing and consumption, controlling for demography and relevant covariates (eg, peer consumption).

► The study explores the association between alcohol marketing and consumption at three levels: overall alcohol consumption, higher-risk drinking in current drinkers and susceptibility in never drinkers.

► The cross-sectional nature of the survey does not enable causal relationships to be drawn about the link between alcohol marketing and either consumption or susceptibility.

**Conclusions** Young people, above and below the legal purchasing age, are aware of a range of alcohol marketing and almost one in five own alcohol branded merchandise. In current drinkers, alcohol marketing awareness was associated with increased consumption and greater likelihood of higher-risk consumption. In never drinkers, ownership of branded merchandise was associated with susceptibility.

## INTRODUCTION

Adolescents and young adults (hereafter 'young people', aged 11–19 years) are a focal population for alcohol research because consumption at this stage of development is associated with increased drinking and risk of concomitant harms in later adulthood.[1 2] Global estimates indicate that consumption by young people is particularly high in Europe, where the proportion of current drinkers (69.5%) is higher than the five other global regions, and the proportion of lifetime abstainers is lower (15.9%).[3] In England, it is estimated that approximately half of children aged 11–15 years (44%) have consumed an alcoholic drink, 1 in 10 have consumed in the past week and 9% have been drunk in the past month.[4] Similar estimates are reported in Scotland and Wales.[5 6] Understanding the drivers of alcohol consumption in young people is important given the immediate and long-term individual, social and economic consequences associated with higher-risk drinking.[7]

One factor routinely cited as shaping alcohol-related attitudes and behaviours in young people is marketing.[8 9] Marketing is fundamentally important to alcohol producers. It represents the primary method of communicating with new and existing consumers, can directly encourage sales and can increase brand salience over competitors. Accordingly, alcohol companies have used highly visible marketing for over 100 years,[10] with the current UK landscape characterised by a complex network of mass media marketing (eg, television), alternative marketing (eg, sponsorship), consumer marketing (eg, price) and stakeholder marketing (eg, to retailers).[9] The importance of marketing to the alcohol industry is evidenced through their annual investment, with Diageo's global marketing expenditure approximately £1.8 billion.[11] Continued consolidation in the alcohol industry has also seen the market become dominated by a small number of transnational producers, creating larger marketing budgets, economies of scale and intense competition.[12]

Content analysis research, which focuses on the marketing output as the unit of analysis, consistently reports that marketing may reach and influence young people. For example, marketing has been reported in media environments where young people may be exposed, including sports,[13] social media,[14] print media[15] and onscreen.[16 17] Content research has also found that marketing may appeal to young people through creative designs, use of topical and real-world associations which may resonate with younger audiences, and by promoting positive connotations around consumption (eg, sociability or desirable lifestyles).[18 19] It has also been suggested that commercial marketing contains ambiguous messages about lower-risk consumption.[20 21]

Systematic reviews of consumer research, which focus on the individual as the unit of analysis, provide consistent evidence that awareness of, and participation with, marketing has a causal influence on young people's consumption, including initiation and frequency of drinking.[22 23] Qualitative research has also suggested that this relationship is more complex than an 'exposure equals consumption' hypothesis, and that young people consider alcohol marketing and branding to hold rich cultural, social and symbolic meaning.[14 24 25] Accordingly, message interpretation research has attempted to move the debate on from whether marketing is associated with consumption and onto how this influence occurs, by identifying psychological mechanisms which mediate the relationship between exposure and consumption.[26 27]

In the UK, the influence of alcohol marketing on young people has been a topic of debate for decades.[9 28] These debates are further supplemented by concerns about the efficacy and effectiveness of self-regulation, the predominant approach employed to control alcohol marketing in the UK. This includes suggestions that self-regulation provides inadequate restrictions, is not consistently enforced or complied with, is retrospective and slow to react to complaints, lacks meaningful sanctions and lags behind modern marketing methods.[9 28–32] There are, however, unresolved issues which have inhibited attempts to move the debate forward. In the UK, the last large-scale assessment of young people's awareness of alcohol marketing is a decade old, was only conducted in Scotland, only sampled adolescents under the minimum purchase age, only considered overall marketing awareness (not frequency) and did not consider higher-risk consumption.[33 34]

In this study, we explore frequency of awareness for alcohol marketing and ownership of alcohol branded merchandise in a demographically representative sample of young people in the UK, including those above and below the legal purchasing age. We also consider what association (if any) awareness of alcohol marketing and ownership of branded merchandise has with alcohol consumption and higher-risk drinking in current drinkers, and susceptibility to drink in never drinkers.

## METHOD

### Design and sample

Data come from the 2017 Youth Alcohol Policy Survey, an online cross-sectional survey conducted with young people aged 11–19 years in the UK (n=3399). Responses were collected during April–May 2017. The survey was hosted by YouGov, a market research company, that recruited a sample intended to be representative of the UK population from their UK panel.[35] Participants aged 16 years or over were approached directly to participate, while those aged under 16 years were approached through existing adult panel members known to have children. A survey weight was provided for each respondent (based on age, gender, ethnicity, region and social grade) to enable descriptive data to be representative of the UK population. The study design was informed by previous cross-sectional surveys in the UK that have explored young people's experiences of alcohol and tobacco marketing.[33 36]

## Measures

### Demography

Alcohol consumption is not homogeneous among young people in the UK.[4–7] It is therefore important to adjust for demographic variation when examining any factors purported to be associated with consumption. In this study, age, gender, ethnicity, resident country (England, Scotland, Wales, Northern Ireland), living status, employment status, educational status, legal purchasing status for alcohol (≥18 years old) and indices of deprivation (IMD), were obtained from information held about panel respondents or survey questions.

### Awareness of alcohol marketing

Awareness of alcohol marketing was assessed through structured, self-reported recall, a method frequently used in consumer research.[33] Participants were prompted with the statement '*Over the last month, how often, if at all, have you seen…*' and then presented with descriptions of nine examples of alcohol marketing: (1) Newspapers or magazines. (2) Television. (3) Billboards. (4) Radio. (5) Adverts on social media (eg, YouTube, Tumblr, Facebook, SnapChat, Instagram or other social media). (6) Famous people in films, music videos or TV or pictured in magazines with alcohol (celebrity endorsement). (7) Sports, games or events sponsorship. (8) Special price offers. (9) Competitions or prize draws. As per recent research,[37 38] a Likert Scale was used to measure frequency of noticing

marketing in the past month for each of the nine examples (1=everyday–6=not in the past month; not sure).

In the UK, survey research which has measured awareness of alcohol marketing has typically used dichotomous response options for each channel (eg, yes/no) and used a summation across these to estimate overall awareness.[33 39] This method, however, only provides insight into the breadth of marketing awareness across channels, not frequency or volume, and therefore lacks sensitivity and may underestimate awareness. To enhance accuracy in this study, the self-reported frequency of awareness for each marketing example was converted into the estimated number of days that marketing had been seen in a 4-week period (ie, '1 month'). This timeframe is consistent with previous research[40 41] and is representative of the minimum number of days in any month. For example, an answer of 'everyday' equated to 28 instances of awareness over 4 weeks (ie, 7 days per week multiplied by 4) and once to twice per week equated to six instances over 4 weeks (ie, 1.5 times per week multiplied by 4) (see table 1 for other response options). Scores across the nine channels were summed to create an aggregate score, providing an approximation of total alcohol marketing awareness in the past month. Estimating total volume of awareness, as opposed to breadth across channels, is consistent with other recent alcohol marketing research.[42 43]

In this study, an aggregate awareness score was only computed when a valid answer had been given for all

| Table 1 | Awareness of alcohol marketing in the past month for young people in the UK | | | | | | | | |
|---|---|---|---|---|---|---|---|---|---|
| **Marketing channel** | **Everyday (28)*** | **Five to six times per week (22)*** | **Three to four times per week (14)*** | **Once to twice per week (6)*** | **Less than once a week (2)*** | **Not in the last month (0)*** | **Not sure** | **Seen least week** | **Median score (IQR)†** |
| | % | % | % | % | % | % | % | % | |
| Adverts for alcohol | | | | | | | | | |
| in newspapers or magazines | 1.9 | 1.8 | 4.9 | 10.2 | 12.2 | 42.3 | 26.8 | 18.8 | 0 (6) |
| on television | 5.0 | 5.4 | 12.0 | 20.5 | 15.4 | 22.4 | 19.3 | 42.9 | 6 (14) |
| on billboards | 3.0 | 3.1 | 7.4 | 14.3 | 17.2 | 30.2 | 24.8 | 27.9 | 2 (6) |
| on radio | 1.1 | 1.0 | 2.3 | 5.1 | 7.1 | 54.7 | 28.8 | 9.4 | 0 (0) |
| on YouTube, Tumblr, Facebook, Snapchat, Instagram or other social media | 2.9 | 2.3 | 8.1 | 14.0 | 15.6 | 32.1 | 25.0 | 27.3 | 2 (6) |
| Famous people in films, music videos, on TV or pictured in magazines with alcohol | 4.9 | 5.3 | 10.8 | 17.6 | 14.4 | 23.2 | 23.6 | 38.7 | 6 (14) |
| Sport sponsorship | 2.4 | 3.4 | 7.9 | 17.0 | 17.4 | 27.8 | 24.1 | 30.7 | 2 (6) |
| Special offers | 5.3 | 5.3 | 12.3 | 18.8 | 14.1 | 21.3 | 22.8 | 41.7 | 6 (14) |
| Competitions | 1.4 | 1.2 | 2.8 | 8.2 | 11.9 | 45.6 | 29.0 | 13.6 | 0 (2) |

*Score for estimating the approximate number of days alcohol marketing was noticed in a 1-month period.
†Median number of alcohol marketing instances noticed in a 1-month period.
Base, all respondents (n=3399): weighted.

nine marketing examples. To provide meaningful interpretative utility, the aggregate score for the valid sample was split into tertiles of low (aggregate score ≤16; awareness approximately every other day), medium (17–53; awareness approximately daily) and high awareness (≥54; awareness almost twice daily). If a participant answered 'not sure' to any of the nine channels they were coded as 'not stated' for the aggregate score. Indicating 'not sure' meant that a respondent's potential aggregate score was, by default, more conservative than those who provided a valid answer to all nine examples. These respondents were therefore coded as a separate 'not sure' category to avoid biasing the proportion of valid respondents considered to have low or medium awareness, or what the tertiles boundaries were.

### Ownership of alcohol brand merchandise
Ownership of alcohol branded merchandise was measured through a single item adapted from previous research.[33 44] Participants were asked '*Do you own any merchandise (such as clothing or drinks glasses) that show an alcoholic drink brand or logo?*' (yes/no/not sure).

### Alcohol consumption status
Participants were asked '*Have you ever had a whole alcoholic drink? Not just a sip?*'.[33 34] Those who answered 'No' were classed as never drinkers while those who answered 'Yes' were classed as ever drinkers.

### Alcohol consumption and higher-risk drinking
Alcohol use was measured through the Alcohol Use Disorders Identification Test–Consumption (AUDIT-C), which assessed frequency of consumption, units drunk in a typical drinking occasion and frequency of heavy episodic drinking. Responses were provided on 5-point scales, with the answers for each item relative to frequency (0=never– 4=four times or more a week), units drunk (0=1–2 units–4=10 units or more) or frequency of heavy episodic drinking (0=never–4=daily or almost daily). Heavy episodic drinking was classified as eight or more units in a single occasion for males, and six or more units for females (1 unit=8 g or 10 mL of alcohol). A diagram depicting the unit content of alcoholic drinks was included to assist calculation of units. Those who answered anything other than 'never' on the first AUDIT-C item were classed as current drinkers and asked to complete the final two items. All other respondents (ie, those stating 'never' for frequency of consumption) were classified as non-drinkers and were not asked to complete the final two items. In current drinkers, a total AUDIT-C score was computed by summing the three AUDIT-C items (0–12), with a cut-off of ≥5 used to identify higher-risk consumption.[45]

### Susceptibility
As per tobacco research, susceptibility was defined as the absence of a firm decision not to drink alcohol in the next year.[36] Never drinkers were classified as 'non-susceptible' if they answered 'definitely no' to the question '*Do you think you will drink alcohol at any time during the next year*?'. Those who answered anything other than 'Definitely no' were classified as susceptible.

### Confounding variables
Confounding factors, reported to influence consumption in young people and used in previous alcohol marketing research, were included as covariates to contextualise any association between marketing and consumption.[33 34 46 47] Frequency of consumption was measured for the mother (female carer), father (male carer) and closest friend (each scored: 1=never–9=every day or almost every day; prefer not to say; not applicable). For all three groups, consumption was collapsed into five categories (never, less than monthly, monthly or fortnightly, at least weekly and not stated). Perceived acceptability of consumption was measured for parents and peers (each scored: 1=totally acceptable–5=totally unacceptable). For both groups, acceptability was converted into dichotomous categories ('Neutral or unacceptable' and 'Acceptable'). For ever drinkers, age of first drink was also measured (≤8 years old–19 years old; can't remember; prefer not to say). Answers were converted into three categories (≤13 years; 14–15 years; >16 years).

### Ethics
YouGov included a lead for ethical and quality assurance, including consent, postsurvey debriefing and signposting to support organisations, and confidentiality and anonymity.

### Patient and public involvement
The survey was developed following cognitive testing with a small sample (n=100) of young people to ensure age and cultural comprehension of the questions. Beyond this, no other patient or public involvement was undertaken.

### Analysis
Data were analysed using SPSS V.23 (SPSS, Chicago, Illinois, USA). Descriptive data were weighted so that percentages and median scores were representative of the demographic profile of the UK population. Bivariate analyses, using $\chi^2$ tests, examined differences in level of alcohol marketing awareness and ownership of branded merchandise between the demographic and confounding variables.

A multivariate linear regression was conducted with current drinkers' AUDIT-C score as the dependent variable (0–12) and awareness of marketing and ownership of branded merchandise as the key independent variables. The following demographic and confounding variables were also included in initial blocks: age; gender; ethnicity; IMD quintile; resident country; educational status; working status; living status; frequency of mother (female carer), father (male carer) and close friend drinking; perceived parental and peer acceptability of consumption; and age at first drink. Categorical variables with at least three categories were converted into dummy (binary) variables to aid interpretation and comparison.

The omitted dummy variable formed the reference category. For example, marketing awareness was a categorical variable with four levels: low, medium, high and not stated. Four binary variables were computed: low awareness, medium awareness, high awareness and not stated (each coded yes=1, no=0). By including medium, high and not stated in the multivariate analysis, and omitting low awareness, the reference category was low awareness. The multivariate regressions therefore demonstrate the extent to which medium or high marketing awareness, relative to low awareness, was associated with alcohol consumption. Reference categories for each variable are displayed in the results.

Two multivariate logistic regressions were conducted with higher-risk drinking (AUDIT-C $\geq$5) among current drinkers and susceptibility to drink among never drinkers as the dependent variables. Marketing awareness and ownership of branded merchandise were the key independent variables. Where applicable, both logistic regressions controlled for the same demographic and confounding variables as the linear regression. Reference categories for categorical independent variables are indicated in the results. For categorical variables which had three or more levels, and were of an ordinal level, the SPSS contrast=difference function enabled comparison of each increasing category level relative to the combined previous category levels. For example, the first comparison with frequency of mother's drinking and higher-risk drinking was 'less than monthly drinking' versus 'mother never drinks', whereas the final comparison was 'at least weekly drinking' versus 'less often'. As the independent variables were categorical, 'not stated' responses were also included as a separate category and compared against the reference category for each variable. This enabled the maximum sample to be retained. For example, the large number of 'not stated' responses on level of marketing awareness could be compared with those for whom marketing awareness could be computed.

All multivariate analyses were conducted on unweighted data as the factors used to construct the weights were included as covariates in the models. The multivariate analyses were repeated on weighted data to check for consistency. As results for the key independent variables (marketing awareness and ownership of branded merchandise) were consistent, only the unweighted results are presented.

## RESULTS
### Sample characteristics
The weighted sample (n=3399) had an average age of 15.18 years (*SD*=2.55; range: 11–19 years), with three quarters (76%) below the legal purchasing age ($\leq$18 years). There was an even distribution for gender (51% male and 49% female). The majority of the sample was white British (76%) and was evenly distributed across IMD (20% in each quintile). Most participants lived in England (84%) with the remainder from Scotland (8%),

Wales (5%) and Northern Ireland (3%). Almost all participants were living at home with parent(s) or other adult family members (90%) and were in some form of education (95%).

### Alcohol consumption and susceptibility
After excluding cases with missing data on drinking status (n=62, weighted), almost half of the weighted sample (48%; n=1590) were current drinkers. Within current drinkers, the average AUDIT-C score was 4.33 (*SD*=2.77). Almost half of current drinkers (44%; n=707) were classified as higher-risk ($\geq$5 AUDIT-C). After excluding cases with missing data on drinking status (n=62, weighted), almost half of the weighted sample (49%; n=1623) was never drinkers. Within never drinkers, half were classified as susceptible (52%; n=841).

### Awareness of alcohol marketing
The most frequent sources of marketing awareness in the past month were adverts on television (median=6 instances per month, IQR=14), celebrity endorsement (median=6, IQR=14) and special offers (median=6, IQR=14) (table 1). More than a third of respondents (range: 39%–43%) had noticed marketing through these channels at least weekly. Billboard adverts (median=2 instances per month, IQR=6), sponsorship (median=2, IQR=6) and social media adverts (median=2, IQR=6) were noticed less than once a week, with at least a quarter of participants (range: 27%–31%) having noticed these at least weekly. Lowest awareness was for adverts in the print press (median=0 instances per month, IQR=6), on radio (median=0, IQR=0) and competitions (median=0, IQR=2). For each marketing example, a fifth or more (range: 19%–29%) were not sure how often, if at all, they had come across alcohol marketing. Overall, 82% had noticed marketing through at least one channel.

### Aggregate alcohol marketing awareness
The median aggregate alcohol marketing awareness score was 32 (IQR=60), equating to noticing 32 instances of alcohol marketing in the past month (under minimum purchase age: median=28; IQR=60). When categorised into tertiles, 35% of the valid sample was classified as having low awareness ($\leq$16 instances per month), 32% had medium awareness (17–53) and 34% had high awareness ($\geq$54). In those under the minimum purchase age, 38% had low awareness, 31% medium and 32% high.

Bivariate $\chi^2$ tests found that higher awareness of alcohol marketing was significantly associated with being male, of legal purchasing age, a current drinker, a higher-risk drinker, not in education, in employment, and perceiving that parents and peers would consider it okay to consume (table 2). Higher awareness was also associated with greater frequency of mother (female carer) consumption, $\chi^2(16)$=38.25, p<0.001, greater frequency of father (male carer) consumption, $\chi^2(16)$=29.55, p<0.05, and greater frequency of close friends drinking,

**Table 2** Classification of alcohol marketing awareness (low, medium and high) by demographic and confounding variables

| Variable | Valid n (n=1411)* | Low awareness (%)† | Medium awareness (%)‡ | High awareness (%)§ | $\chi^2$ | P value |
|---|---|---|---|---|---|---|
| Gender | | | | | 9.26 | <0.01 |
| Male | 735 | 32.1 | 30.5 | 37.4 | | |
| Female | 676 | 37.3 | 32.8 | 29.9 | | |
| Ethnicity | | | | | 1.09 | n.s. |
| White British | 1082 | 34.5 | 32.3 | 33.3 | | |
| Other ethnicity | 317 | 35.0 | 29.3 | 35.6 | | |
| IMD Quintile | | | | | 10.56 | n.s. |
| 1 (most deprived) | 247 | 34.4 | 26.3 | 39.3 | | |
| 2 | 266 | 35.7 | 28.2 | 36.1 | | |
| 3 | 288 | 36.8 | 31.9 | 31.2 | | |
| 4 | 292 | 32.2 | 34.6 | 33.2 | | |
| 5 (least deprived) | 317 | 34.1 | 35.3 | 30.6 | | |
| Country lived in | | | | | 6.89 | n.s. |
| England | 1230 | 34.5 | 32.0 | 33.6 | | |
| Scotland | 93 | 34.4 | 33.3 | 32.3 | | |
| Wales | 53 | 39.6 | 30.2 | 30.2 | | |
| Northern Ireland | 34 | 29.4 | 17.6 | 52.9 | | |
| Legal purchase age | | | | | 14.10 | <0.01 |
| No | 995 | 37.6 | 30.7 | 31.8 | | |
| Yes | 416 | 27.4 | 33.7 | 38.9 | | |
| Current drinker | | | | | 114.04 | <0.001 |
| No | 609 | 49.9 | 26.9 | 23.2 | | |
| Yes | 784 | 23.1 | 34.8 | 42.1 | | |
| Higher risk drinker | | | | | 85.84 | <0.001 |
| No | 1027 | 41.7 | 29.1 | 29.2 | | |
| Yes | 384 | 15.6 | 38.3 | 46.1 | | |
| Education | | | | | 13.90 | <0.001 |
| Not in education | 79 | 17.7 | 31.6 | 50.6 | | |
| In education | 1330 | 35.6 | 31.7 | 32.8 | | |
| Working status | | | | | 7.93 | <0.05 |
| Not in work | 1282 | 35.6 | 31.6 | 32.8 | | |
| In work | 127 | 24.4 | 32.3 | 43.3 | | |
| Parents accept use | | | | | 63.06 | <0.001 |
| No | 722 | 44.2 | 28.4 | 27.4 | | |
| Yes | 689 | 24.4 | 35.0 | 40.6 | | |
| Peer accept use | | | | | | |
| No | 410 | 51.5 | 24.4 | 24.1 | 73.08 | <0.001 |
| Yes | 1001 | 27.7 | 34.6 | 37.8 | | |

Due to a large number of categories, analysis of how awareness of alcohol marketing varied by mother (female carer), father (male carer) and close friend frequency consumption only reported in text.
$\chi^2$ = Bivariate Pearson $\chi^2$.
*Valid sample excludes those who had reported 'not sure' to any marketing channels; sample is weighted.
†Low awareness equals ≤16 instances per month (ie, once every other day).
‡Medium awareness equals 17–53 instances per month (ie, almost once a day or more).
§High awareness equals ≥54 instances per month (ie, almost twice a day or more).

$\chi^2(16)=198.51$, p<0.001. There was no difference in awareness by ethnicity, IMD quintile or resident country.

### Owning alcohol branded merchandise

Almost a fifth of participants (17%) reported owning alcohol branded merchandise. Bivariate $\chi^2$ tests found that ownership of branded merchandise was significantly associated with being of white British ethnicity, of legal purchase age, a current drinker, a higher-risk drinker, not in education, in employment, and perceiving that parents and peers would consider it okay to consume (table 3). Ownership of branded merchandise was

**Table 3** Ownership of alcohol branded items by demographic and confounding variables

| Variable | Valid n (n=3276)* | Own branded merchandise (%) | $\chi^2$ | P value |
|---|---|---|---|---|
| Gender | | | 2.71 | n.s. |
| Male | 1679 | 18.5 | | |
| Female | 1597 | 16.3 | | |
| Ethnicity | | | 16.68 | <0.001 |
| White British | 2506 | 19.0 | | |
| Other ethnicity | 745 | 12.5 | | |
| IMD quintile | | | 15.73 | <0.01 |
| 1 (most deprived) | 652 | 13.5 | | |
| 2 | 646 | 21.1 | | |
| 3 | 644 | 17.2 | | |
| 4 | 662 | 19.5 | | |
| 5 (least deprived) | 655 | 16.0 | | |
| Country lived in | | | 0.97 | n.s. |
| England | 2759 | 17.4 | | |
| Scotland | 260 | 16.2 | | |
| Wales | 155 | 16.8 | | |
| Northern Ireland | 103 | 20.4 | | |
| Legal purchase age | | | 100.33 | <0.001 |
| No | 2488 | 13.7 | | |
| Yes | 788 | 29.2 | | |
| Current drinker | | | 256.07 | <0.001 |
| No | 1683 | 7.2 | | |
| Yes | 1549 | 28.7 | | |
| Higher-risk drinker | | | 222.98 | <0.001 |
| No | 2543 | 12.3 | | |
| Yes | 690 | 36.7 | | |
| Education | | | 43.73 | <0.001 |
| Not in education | 161 | 36.6 | | |
| In education | 3106 | 16.4 | | |
| Working status | | | 31.08 | <0.001 |
| Not in work | 3028 | 16.3 | | |
| In work | 239 | 30.5 | | |
| Parents accept use | | | 189.06 | <0.001 |
| No | 1920 | 9.7 | | |
| Yes | 1357 | 28.2 | | |
| Peer accept use | | | | |
| No | 1066 | 8.0 | 97.68 | <0.001 |
| Yes | 2210 | 21.9 | | |

Due to a large number of categories, analysis of how ownership of alcohol branded merchandise varied by mother (female carer), father (male carer) and close friend frequency consumption only reported in text.
$\chi^2$ = Bivariate Pearson $\chi^2$.
*Valid sample refers to those who answered 'yes' or 'no'. Missing cases due to 'don't know' response (n=123). Sample is weighted.

also associated with greater frequency of mother (female carer) consumption, $\chi^2(8)=44.11$, p<0.001, greater frequency of father (male carer) consumption, $\chi^2(8)=56.49$, p<0.001, and greater frequency of close friends drinking, $\chi^2(8)=178.76$, p<0.001. There was also an overall effect of IMD, $\chi^2(4)=15.73$, p<0.01, although this had no distinct pattern across escalating deprivation. There was no difference by resident country or gender.

### Association between alcohol marketing and AUDIT-C scoring

A multivariate linear regression examined the association between marketing awareness, ownership of branded merchandise and AUDIT-C scoring in current drinkers (table 4). After controlling for demographic and confounding factors, medium alcohol marketing awareness ($b$=0.79, 95% CI 0.37 to 1.21, p<0.001), or high awareness ($b$=0.85, 95% CI 0.44 to 1.26, p<0.001), compared with low awareness, was associated with higher AUDIT-C score, as was ownership of branded merchandise ($b$=0.79, 95% CI 0.55 to 1.04, p<0.001). Of the demographic variables, being older (p<0.001), male (p=0.01), from a more affluent IMD (p<0.01), in education (p<0.01) and living independently of parents or adult family members (p<0.001) was also associated with higher AUDIT-C score in the final model. Of the confounding variables, having a close friend who drinks at least weekly (p<0.001), and perceiving that parents consider it acceptable to consume (p<0.05) was also associated with higher AUDIT-C score. Having a first alcoholic drink at at least 16 years (p<0.001) was associated with lower AUDIT-C score, compared with those who first drank aged 14–15 years.

### Association between alcohol marketing and higher-risk consumption

A multivariate logistic regression examined the association between marketing awareness, ownership of branded merchandise and higher-risk drinking in current drinkers (table 5). After controlling for demographic and confounding factors, medium alcohol marketing awareness (adjusted OR=2.18, 95% CI 1.39 to 3.42, p<0.001), high awareness (adjusted OR (AOR)=1.43, 95% CI 1.01 to 2.02, p<0.05) and owning branded merchandise (AOR=1.71, 95% CI 1.31 to 2.22, p<0.001) were associated with higher-risk drinking. Of the demographic variables, being older (p<0.001), male (p<0.05), from England compared with Wales (p<0.05), in education (p<0.05) and living independently (p<0.05) were associated with higher-risk drinking in the final model. Of the confounding variables, increasing frequency of close friend consumption (p<0.001) and having had first drink at age 14–15 years or younger (p<0.001), was associated with higher-risk consumption.

### Association between alcohol marketing and susceptibility to consume

A multivariate logistic regression examined the association between marketing awareness, ownership of

**Table 4**  Association between alcohol marketing awareness and AUDIT-C scoring in current drinkers

| Variables and reference categories | Unstandardised coefficients | | | | Standard coefficients | | |
|---|---|---|---|---|---|---|---|
| | b | 95% CI lower | 95% CI upper | SE | β | t | P value |
| Constant | −5.57 | −7.05 | −4.09 | 0.75 | | −7.40 | <0.001 |
| Age | 0.43 | 0.35 | 0.51 | 0.04 | 0.30 | 10.70 | <0.001 |
| Gender | | | | | | | |
| Male (vs female) | 0.31 | 0.09 | 0.54 | 0.11 | 0.06 | 2.76 | <0.01 |
| Ethnicity | | | | | | | |
| White British (vs other) | 0.08 | −0.24 | 0.40 | 0.16 | 0.01 | 0.52 | n.s. |
| IMD quintile | | | | | | | |
| (1: most deprived to 5: most affluent) | 0.11 | 0.03 | 0.20 | 0.04 | 0.06 | 2.68 | <0.01 |
| Country | | | | | | | |
| Scotland (vs England) | −0.05 | −0.40 | 0.31 | 0.18 | −0.01 | −0.26 | n.s. |
| Wales and Northern Ireland (vs England) | −0.37 | −0.76 | 0.01 | 0.20 | −0.04 | −1.90 | n.s. |
| Educational status | | | | | | | |
| In education (vs not) | 0.66 | 0.22 | 1.10 | 0.22 | 0.07 | 2.96 | <0.01 |
| Working status | | | | | | | |
| Working (vs not) | 0.31 | −0.06 | 0.67 | 0.19 | 0.04 | 1.66 | n.s. |
| Living status | | | | | | | |
| Living independently (vs with parents/adult family) | 0.87 | 0.54 | 1.20 | 0.17 | 0.12 | 5.17 | <0.001 |
| Not stated (vs with parents/adult family) | 0.42 | −0.66 | 1.49 | 0.55 | 0.02 | 0.76 | n.s. |
| Frequency of mother drinking | | | | | | | |
| Never (vs at least monthly) | 0.04 | −0.41 | 0.49 | 0.23 | 0.00 | 0.17 | n.s. |
| Less than monthly (vs at least monthly) | −0.31 | −0.63 | 0.00 | 0.16 | −0.04 | −1.94 | n.s. |
| Not stated (vs at least monthly) | 0.42 | −0.20 | 1.03 | 0.31 | 0.03 | 1.33 | n.s. |
| Frequency of father drinking | | | | | | | |
| Never (vs at least monthly) | 0.21 | −0.33 | 0.75 | 0.27 | 0.02 | 0.77 | n.s. |
| Less than monthly (vs at least monthly) | 0.32 | −0.08 | 0.72 | 0.20 | 0.03 | 1.57 | n.s. |
| Not stated (vs at least monthly) | 0.33 | −0.04 | 0.71 | 0.19 | 0.04 | 1.76 | n.s. |
| Frequency of close friends drinking | | | | | | | |
| At least weekly (vs less often or never) | 1.44 | 1.19 | 1.69 | 0.13 | 0.26 | 11.32 | <0.001 |
| Not stated (vs less than weekly or never) | −0.49 | −0.85 | −0.12 | 0.19 | −0.06 | −2.61 | <0.01 |
| Parents' views | | | | | | | |
| Drinking acceptable (vs neutral/unacceptable) | 0.29 | 0.01 | 0.57 | 0.14 | 0.05 | 2.06 | <0.05 |
| Peer views | | | | | | | |
| Drinking acceptable (vs neutral/unacceptable) | 0.08 | −0.32 | 0.48 | 0.21 | 0.01 | 0.38 | n.s. |
| Age of first drink | | | | | | | |
| Age 13 years or under (vs 14–15 years) | 0.22 | −0.07 | 0.51 | 0.15 | 0.04 | 1.50 | n.s. |
| Age 16 years or over (vs 14–15 years) | −1.33 | −1.63 | −1.04 | 0.15 | −0.21 | −8.82 | <0.001 |
| Not stated (vs 14–15 years) | −0.48 | −0.89 | −0.07 | 0.21 | −0.05 | −2.28 | <0.05 |
| Alcohol marketing awareness | | | | | | | |
| Medium (vs low awareness) | 0.79 | 0.37 | 1.21 | 0.21 | 0.11 | 3.70 | <0.001 |
| High (vs low awareness) | 0.85 | 0.44 | 1.26 | 0.21 | 0.12 | 4.08 | <0.001 |
| Not stated (vs low awareness) | 0.40 | 0.04 | 0.76 | 0.18 | 0.07 | 2.20 | <0.05 |
| Own alcohol branded merchandise | | | | | | | |
| Yes (vs no/not sure) | 0.79 | 0.55 | 1.04 | 0.13 | 0.13 | 6.30 | <0.001 |

Based on current drinkers: n=1592; data are unweighted.
Model shown is final block. Total variance explained (adjusted $R^2$=0.36). Durbin Watson=2.01.
Final step model change: $F_{(4, 1,564)}$=17.44, p<0.001.
Overall Final model analysis of variance: $F_{(27, 1,564)}$=34.33, p<0.001.
DV, Alcohol Use Disorders Identification Test–Consumption (AUDIT-C) scoring (0–12).

**Table 5** Logistic regression of association between alcohol marketing and higher risk consumption among current drinkers

| | Higher risk consumption among current drinkers | | | | |
| --- | --- | --- | --- | --- | --- |
| | n | AOR* | 95% CI lower | 95% CI upper | P value |
| Age | 1592 | 1.40 | 1.28 | 1.53 | <0.001 |
| Gender | | | | | |
| Female | 824 | Ref | | | |
| Male | 768 | 1.32 | 1.04 | 1.68 | <0.05 |
| Ethnicity | | | | | |
| Other | 228 | Ref | | | |
| White British | 1364 | 0.97 | 0.69 | 1.37 | n.s. |
| IMD quintile | | | | | n.s. |
| 1 (most deprived) | 232 | Ref | | | |
| 2 vs 1 | 334 | 1.65 | 1.08 | 2.52 | <0.05 |
| 3 vs 1,2 | 324 | 1.26 | 0.90 | 1.76 | n.s. |
| 4 vs 1,2,3 | 340 | 1.21 | 0.90 | 1.64 | n.s. |
| 5 (most affluent) vs 1,2,3,4 | 362 | 1.23 | 0.93 | 1.64 | n.s. |
| Country | | | | | n.s. |
| England | 1243 | Ref | | | |
| Scotland | 197 | 0.88 | 0.60 | 1.28 | n.s. |
| Wales | 116 | 0.58 | 0.36 | 0.93 | <0.05 |
| Northern Ireland | 36 | 1.35 | 0.60 | 3.01 | n.s. |
| Educational status | | | | | |
| Not in education | 146 | Ref | | | |
| In education | 1446 | 1.61 | 1.01 | 2.55 | <0.05 |
| Working status | | | | | |
| Not working | 1374 | Ref | | | |
| Working (full-time or part-time) | 218 | 1.43 | 0.97 | 2.09 | n.s. |
| Living status | | | | | |
| Living with parents/ adult family | 1307 | Ref | | | |
| Living independently | 268 | 1.56 | 1.09 | 2.23 | <0.05 |
| Not stated | 17 | 1.58 | 0.54 | 4.60 | n.s. |
| Frequency of mother drinking | | | | | <0.05 |
| Never | 115 | Ref | | | |
| Less than monthly versus never | 284 | 0.47 | 0.27 | 0.79 | <0.01 |
| Monthly or fortnightly versus less often | 279 | 1.22 | 0.83 | 1.79 | n.s. |
| At least weekly versus less often | 849 | .93 | .70 | 1.24 | n.s. |
| Not stated versus all other categories | 65 | 1.50 | .78 | 2.88 | n.s. |
| Frequency of father drinking | | | | | n.s. |
| Never | 76 | Ref | | | |
| Less than monthly versus never | 160 | 1.40 | 0.72 | 2.73 | n.s. |
| Monthly or fortnightly versus less often | 201 | 0.73 | 0.46 | 1.19 | n.s. |
| At least weekly versus less often | 964 | 0.83 | 0.61 | 1.15 | n.s. |
| Not stated versus all other categories | 191 | 1.14 | 0.75 | 1.72 | n.s. |

Continued

**Table 5** Continued

| | n | AOR* | 95% CI lower | 95% CI upper | P value |
|---|---|---|---|---|---|
| **Higher risk consumption among current drinkers** | | | | | |
| Frequency of close friends drinking | | | | | <0.001 |
| Never | 72 | Ref | | | |
| Less than monthly versus never | 187 | 0.68 | 0.32 | 1.42 | n.s. |
| Monthly or fortnightly versus less often | 463 | 2.20 | 1.44 | 3.35 | <0.001 |
| At least weekly versus less often | 667 | 3.41 | 2.48 | 4.70 | <0.001 |
| Not stated versus all other categories | 203 | 0.57 | 0.37 | 0.89 | 0.013 |
| Parents' views | | | | | |
| Neutral or unacceptable | 473 | Ref | | | |
| Drinking acceptable | 1119 | 0.92 | 0.68 | 1.24 | n.s. |
| Peer views | | | | | |
| Neutral or unacceptable | 156 | Ref | | | |
| Drinking acceptable | 1436 | 1.41 | 0.88 | 2.25 | n.s. |
| Age of first drink | | | | | <0.001 |
| Age 13 years or under | 472 | Ref | | | |
| Age 14–15 years (vs 13 years or under) | 535 | 0.86 | 0.63 | 1.18 | n.s. |
| Age 16 years or over (vs younger) | 412 | 0.26 | 0.19 | 0.35 | <0.001 |
| Not stated | 173 | 0.89 | 0.59 | 1.35 | n.s. |
| Alcohol marketing awareness | | | | | <0.001 |
| Low awareness | 184 | Ref | | | |
| Medium versus low | 274 | 2.18 | 1.39 | 3.42 | <0.001 |
| High versus medium and low | 326 | 1.43 | 1.01 | 2.02 | <0.05 |
| Not stated versus all other categories | 808 | 0.85 | 0.67 | 1.08 | n.s. |
| Own alcohol branded merchandise | | | | | |
| No or not sure | 1138 | Ref | | | |
| Yes | 454 | 1.71 | 1.31 | 2.22 | <0.001 |

Based on current drinkers (n=1592); data are unweighted.
Test of model coefficients in final block: $\chi^2$ (35)=477.29, p<0.001.
Hosmer-Lemeshow test for final block $\chi^2$ (8)=11.66, p=0.17.
Nagelkerke's $R^2$ for final block=0.35.
Cases correctly classified in final block: 72% in final block.
*Adjusted for all other variables in the model.
AOR, adjusted OR; Ref, reference category.
DV, higher-risk drinking on the Alcohol Use Disorders Identification Test–Consumption (AUDIT-C) (>5), 1=higher risk (n=699) and 0=lower risk (n=893).

branded merchandise and susceptibility to drink in never drinkers (table 6). After controlling for demographic and confounding variables, awareness of alcohol marketing was not associated with susceptibility, but ownership of branded merchandise was, with those who owned branded merchandise almost twice as likely to be susceptible compared with those who did not (AOR=1.98, 95% CI 1.20 to 3.24, p<0.01). Of the demographic variables, only being white British (p<0.01) was associated with susceptibility in the final model. Of the confounding variables, frequency of mother (female carer) consumption (p<0.001), frequency of father (male carer) consumption (p<0.05), frequency of close friend consumption (p<0.001) and perceived peer approval (p<0.001) were associated with susceptibility.

## DISCUSSION

The findings indicate that young people in the UK are aware of a variety of alcohol marketing and almost a fifth own branded merchandise. The results also show that awareness of marketing and ownership of branded

**Table 6**  Logistic regression of association between alcohol marketing and never drinkers' susceptibility to drink

| | Susceptibility to drink among never drinkers | | | | |
|---|---|---|---|---|---|
| | n | AOR* | 95% CI lower | 95% CI upper | P value |
| Age | 1580 | 1.05 | 0.98 | 1.13 | n.s. |
| Gender | | | | | |
| Female | 791 | Ref | | | |
| Male | 789 | 1.09 | 0.88 | 1.37 | n.s. |
| Ethnicity | | | | | |
| Other | 377 | Ref | | | |
| White British | 1203 | 1.51 | 1.12 | 2.03 | <0.01 |
| IMD quintile | | | | | n.s. |
| 1 (most deprived) | 399 | Ref | | | |
| 2 vs 1 | 278 | 1.13 | 0.80 | 1.60 | n.s. |
| 3 vs 1,2 | 355 | 1.02 | 0.76 | 1.36 | n.s. |
| 4 vs 1,2,3 | 233 | 0.88 | 0.64 | 1.22 | n.s. |
| 5 (most affluent) vs 1,2,3,4 | 315 | 0.84 | 0.63 | 1.11 | n.s. |
| Country | | | | | n.s. |
| England | 1193 | Ref | | | |
| Scotland | 191 | 1.14 | 0.80 | 1.61 | n.s. |
| Wales | 115 | 1.09 | 0.70 | 1.69 | n.s. |
| Northern Ireland | 81 | 0.96 | 0.58 | 1.59 | n.s. |
| Educational status | | | | | |
| Not in education | 25 | Ref | | | |
| In education | 1555 | 0.67 | 0.20 | 2.25 | n.s. |
| Working status | | | | | |
| Not working | 1550 | Ref | | | |
| Working (full-time or part-time) | 30 | 2.59 | 0.83 | 8.11 | n.s. |
| Living status | | | | | n.s. |
| Living with parents/ adult family | 1545 | Ref | | | |
| Living independently | 28 | 0.51 | 0.20 | 1.28 | n.s. |
| Not stated | 7 | 1.57 | 0.27 | 9.11 | n.s. |
| Frequency of mother drinking | | | | | <0.001 |
| Never | 321 | Ref | | | |
| Less than monthly versus never | 382 | 2.38 | 1.58 | 3.59 | <0.001 |
| Monthly or fortnightly versus less often | 242 | 1.66 | 1.15 | 2.39 | <0.01 |
| At least weekly versus less often | 560 | 1.47 | 1.11 | 1.94 | <0.01 |
| Not stated versus all other categories | 75 | 1.25 | 0.70 | 2.25 | n.s. |
| Frequency of father drinking | | | | | <0.05 |
| Never | 273 | Ref | | | |
| Less than monthly versus never | 217 | 1.88 | 1.17 | 3.01 | <0.01 |
| Monthly or fortnightly versus less often | 232 | 1.11 | 0.75 | 1.64 | n.s. |
| At least weekly versus less often | 686 | 1.39 | 1.05 | 1.84 | <0.05 |
| Not stated versus all other categories | 172 | 1.06 | 0.71 | 1.58 | n.s. |

Continued

**Table 6** Continued

| | | Susceptibility to drink among never drinkers | | | |
|---|---|---|---|---|---|
| | n | AOR* | 95% CI lower | 95% CI upper | P value |
| Frequency of close friends drinking | | | | | <0.001 |
| Never | 922 | Ref | | | |
| Less than monthly versus never | 162 | 3.46 | 2.26 | 5.27 | <0.001 |
| Monthly or fortnightly versus less often | 80 | 3.32 | 1.66 | 6.65 | <0.001 |
| At least weekly versus less often | 83 | 0.70 | 0.39 | 1.26 | n.s. |
| Not stated versus all other categories | 333 | 0.61 | 0.43 | 0.86 | <0.01 |
| Parents' views | | | | | |
| Neutral or unacceptable | 1364 | Ref | | | |
| Drinking acceptable | 216 | 1.00 | 0.70 | 1.44 | n.s. |
| Peer views | | | | | |
| Neutral or unacceptable | 894 | Ref | | | |
| Drinking acceptable | 686 | 2.29 | 1.77 | 2.96 | <0.001 |
| Alcohol marketing awareness | | | | | n.s. |
| Low awareness | 279 | Ref | | | |
| Medium versus low | 148 | 1.44 | 0.92 | 2.28 | n.s. |
| High versus medium and low | 117 | 1.16 | 0.71 | 1.90 | n.s. |
| Not stated versus all other categories | 1036 | 1.21 | 0.94 | 1.56 | n.s. |
| Own alcohol branded merchandise | | | | | |
| No or not sure | 1476 | Ref | | | |
| Yes | 104 | 1.98 | 1.20 | 3.24 | <0.01 |

Based on never drinkers (n=1580) data are unweighted.
Test of model coefficients in final block: $\chi^2$ (32)=337.46, p<0.001.
Hosmer-Lemeshow test for final block $\chi^2$ (8)=5.86, p=0.663
Nagelkerke's $R^2$ for final block=0.26.
Cases correctly classified in final block: 69%
*Adjusted for all other variables in the model,
AOR, adjusted OR; Ref, reference category.
Dedpendent variable, susceptibility: 1=susceptible (n=830)=0; not susceptible (n=750).

merchandise are associated with increased consumption and higher-risk drinking in current drinkers, and that ownership of branded merchandise is associated with susceptibility in never drinkers. We address key evidence gaps in the UK by exploring frequency of marketing awareness (not just breadth of exposure) and demonstrating an association between marketing and both consumption and susceptibility in young people above and below the legal purchasing age from across the UK.

The findings are consistent with suggestions that alcohol marketing appears in contexts which may reach young people, those under the legal purchasing age.[8 9] Awareness included mass media marketing (eg, television), alternative marketing (eg, sponsorship and celebrity endorsement), consumer marketing (eg, price offers) and digital media. This highlights the dynamic nature of '360-degree' marketing strategies and how they reach young people in offline and online environments.[9 48] The

results extend understanding by showing how frequently young people see alcohol marketing; with at least 1 in 10 reporting daily or almost daily awareness through three of the nine marketing examples. Approximately half of the sample had seen at least 32 instances of alcohol marketing per month, which equates to awareness at least once a day. Although there were expected differences in awareness between drinkers and never drinkers,[33 49] there were no differences between key demographic groups, including ethnicity, indices of deprivation and resident country. This suggests that awareness of alcohol marketing occurs in young people across the UK, and is not isolated to a minority of demographic groups.

The results are consistent with longitudinal research which has shown a link between marketing and increased consumption in young people.[22 23 34 38] Although marketing awareness did not have an association with susceptibility in never drinkers, ownership of branded

merchandise did. Research has reported that participation with marketing has a stronger association with consumption than awareness.[33 34 39 49] Our findings therefore suggest that the effect of participation is pronounced in never drinkers. Nevertheless, as research suggests that not all alcohol marketing or brands are equally appealing to youth,[25 50] it is possible that focusing on aggregated alcohol marketing awareness (the approach in this study) may have disguised associations between individual examples of marketing and susceptibility in never drinkers. The findings also extend understanding by showing an association between marketing and consumption across young adulthood. This includes an association with susceptibility and consumption in young people under the legal purchasing age and higher-risk drinking in newly legal drinkers. Newly legal drinkers are an important target for alcohol marketers[18] and are a key under-researched group.[51] The findings therefore highlight the importance of considering the wider role that marketing plays on consumption, not just in those under the purchasing age.

Except for the Scottish government's decision to implement minimum unit pricing,[52] there has been little recent change in UK's self-regulatory and co-regulatory frameworks for alcohol marketing.[53 54] It is claimed that such self-regulatory approaches provide inadequate restrictions, are not suitably enforced, are retrospective and slow to react to complaints, and lack meaningful sanctions.[9 28–32] Although statutory regulations are cited as an alternative approach,[28] studies have also questioned whether current examples, such as the Loi Évin in France, are being enforced properly or whether they reduce marketing exposure.[13 37] Further research exploring the perspectives of stakeholders involved in the production, research, consumption and regulation of marketing would be of value to identify feasible and effective options to reduce youth exposure and form a consensus on appropriate action.[55 56]

There are limitations. First, the cross-sectional design cannot identify a causal relationship between marketing and consumption, although a directional effect is supported by longitudinal research.[22 23] Moreover, that marketing had any association with consumption and susceptibility at all suggests that it must at least play either an initiating or reinforcing role. Second, the results are only partially representative of young adults above the legal purchasing age, although other research has shown similar trends in older young adults.[39] Third, the marketing channels measured are not exhaustive and, consequently, the results may underestimate awareness. Examples of omitted marketing include packaging, cinema, product placement and a broader range of digital marketing.[33 39 49] It was also not possible to decipher whether 'not sure' responses indicated uncertainty over whether a participant had seen alcohol marketed at all through a channel or uncertainty on the frequency of awareness. This influenced the design of the regression models (to account for a 'Not sure' category). Third, except for owning branded merchandise, the study only measured awareness of marketing, but not participation. As participation is reported to have a stronger effect,[34 49] the results may underestimate the association between marketing and drinking outcomes. Finally, measurement of owning branded merchandise also only included two examples as prompts (clothing and drinks glasses). It is possible that different prompts may have altered recall, and that multiple items or a free-text response option would have provided greater clarity on merchandise owned.

## CONCLUSION

This paper makes important contributions to understanding by exploring awareness of alcohol marketing and ownership of branded merchandise by young people from across the UK, three quarters of who were under the legal purchasing age. The results highlight that '360-degree' marketing strategies have created several avenues for young people to be exposed to, or involved with, alcohol marketing, and that this is associated with consumption and higher-risk drinking in current drinkers and susceptibility in never drinkers. Further scrutiny and examination of the UK's self-regulatory approach and viable alternatives are needed to identify feasible, appropriate and effective means of reducing marketing exposure in young people.

**Acknowledgements** The authors thank YouGov for their assistance in preparing and managing delivery of the survey and data. The authors also thank Gillian Rosenberg from the CPRC at Cancer Research UK for her role in managing the project and providing information on YouGov fieldwork. The authors also thank NatCen Social Research for their contribution to the original scoping studies used in the development of this project.

**Contributors** LH, CT and JV led the study design and data acquisition. LH, CT, JV were involved in design of the study tools with support from AMM. NC and AMM planned the analysis, and this was conducted by AMM. NC and AMM led interpretation of the results, with input from LH, CT and JV. NC drafted the manuscript, with support from AMM on methods and results, and all authors provided feedback and approved the final version of the manuscript.

**Funding** This work was supported by a grant from Cancer Research UK (1107098).

**Competing interests** NC is a board member of Alcohol Focus Scotland. All other authors have no conflict of interest to declare.

**Patient consent for publication** Not required.

**Ethics approval** University of Stirling's General University Ethics Panel (GUEP59).

**Provenance and peer review** Not commissioned; externally peer reviewed.

**Data sharing statement** The raw data (deindentified participant survey responses) from the Youth Alcohol Policy are held by the Cancer Policy Research Centre at Cancer Research UK.

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
