## [Reviewer comments · BMJ Open]

ARTICLE DETAILS

TITLE (PROVISIONAL)	Awareness of alcohol marketing, ownership of alcohol branded merchandise, and the association with alcohol consumption, higher-risk drinking, and drinking susceptibility in adolescents and young adults: A cross-sectional survey in the United Kingdom.
AUTHORS	Critchlow, Nathan; Mackintosh, Anne Marie; Thomas, Christopher; Hooper, Lucie; Vohra, Jyotsna

VERSION 1 – REVIEW

REVIEWER	Jonathan Noel Assistant Professor Johnson & Wales University, USA
REVIEW RETURNED	23-Jul-2018

GENERAL COMMENTS	This manuscript seeks to update the literature on awareness and effects of alcohol marketing in the U.K., with a particular emphasis on the association of awareness of marketing and alcohol consumption. Overall, this is a well written manuscript that will add to the current literature. Specific questions and comments to improve the manuscript are below. Introduction: 1) Paragraph 1, sentence 1 - Age ranges would be useful here because of the intention to use "young people" as a general term for both adolescents and young adults.2) Last paragraph - A description of the U.K. regulatory environment would be useful in the Introduction to better orient the reader. This should include any literature demonstrating the effectiveness or ineffectiveness of the systems used. Methods: 1) Design and Sample, sentence 1 - Does the age range of 11-19 year olds correspond to the authors' definition of adolescents and young adults?2) Demography - What was the purpose of collecting each of these variables? Why are they important to adjust for in the final analysis?3) Awareness of alcohol marketing, paragraph 2 - It would be useful to provide an example of how awareness was operationalized for lower exposure than the "everyday" category.4) Awareness of alcohol marketing, paragraph 2 - Several studies have aggregated values across different media to determine awareness of alcohol marketing. Did the authors' use a previous validated aggregation approach (which would require citations) or was a method created specifically for the study (which would require further explanation about why this method is reliable and valid)?
--

	5) Awareness of alcohol marketing, paragraph 2, sentence 4 - What is the rationale for only including cases where there was a valid answer for all channels? The values from these participants would most likely be a conservative measure of exposure and are unlikely to bias the results towards a statistically significant finding. I suggest conducting a sensitivity analysis including these cases to determine the effects on the results. 6) Awareness of alcohol marketing, paragraph 2, last sentence - What was the rationale for using tertiles to categorize marketing awareness? 7) Ownership - Where did the question on ownership of alcohol brand merchandise originate from? Has it been validated previously? 8) Drinking status - Where did the drinking status questions originate from? Have they been validated previously? 9) Alcohol consumption - Was the AUDIT-C given to all participants or only those that were considered lifetime drinkers? 10) Confounding variables - I agree that these are potential confounders, but it is important to include citations that conclude these are important factors in alcohol consumption. 11) Confounding variables, sentence 3 - What is the justification or who promoted the idea of converting self-reported consumption into number of days alcohol was consumed each year? That appears to have the potential to inflate alcohol consumption. 12) Analysis, sentence 1 - Please include the corporate information for SPSS. 13) Analysis, sentence 2 - If this was a representative sample, what is the purpose of weighting the data? What characteristics factored into the weights? Was a sensitivity analysis conducted with and without the weights to determine the impact? 14) Analysis, paragraph 2, sentence 1 - This is not an appropriate rationale for conducting an analysis on unweighted data. The weighting procedure prior to analysis and the adjustment of covariates within a statistical analysis are inherently different computations that may or may not be related to each other. I suggest conducting the analysis using the weighted data for consistency purposes. 15) Analysis, paragraph 2, sentence 2 - It may be better to state "multivariable linear regression." Upon reading the term "hierarchical," my first reaction was to think about multi-level modeling. It was a quick over thinking on my part but just double check these terms to ensure accuracy and comprehension. 16) Analysis, paragraph 3, sentence 4 - State the reference categories in the methods. Results: 1) Owning alcohol branded merchandise - After all the detail of the previous section, this part seems to be lacking. Were there any differences across demographic categories? 2) Throughout the Methods, the results of alcohol marketing awareness and ownership of branded merchandise should be positioned at the top of each section, not at the bottom. Those variables are the reason for conducting the study. highlight them. 3) Refrain from repeating the bulk of the information from the tables. Be concise and only highlight information stemming from the study hypotheses or novel findings. Discussion:
--	---

	1) Re-think phrasing and sentence structure in a few locations. Several sentences start with "That" and I often had to re-read a sentence to fully comprehend its meaning. 2) Paragraph 5 - It would be useful to better differentiate between the limitations of the current study and future research studies that could be conducted. I understand the relationship but they are different entities. Tables: 1) I suggest placing the results for marketing awareness and alcohol branded merchandise at the top of the tables.
--	--

REVIEWER	Alisa Padon, Research Scientist Public Health Institute, USA
REVIEW RETURNED	08-Aug-2018

GENERAL COMMENTS	The study explores the association between alcohol marketing and consumption at three levels: overall alcohol consumption, higher-risk consumption, and susceptibility in never-drinkers. They make a strong case for the necessity of this research, given the length of time since the last large-scale assessment of young people's awareness of alcohol marketing in the UK, and its limitations. The sample size is robust. I did have a number of questions largely related to clarifying measures used, but overall the study is an important addition to the literature. 1. Was there any eligibility criteria? What was the overall response rate? Was parental permission required? 2. p. 7. Are the channels mutually exclusive? Does this "channel": "(6) famous people in films, music videos or TV or pictured in magazines;" mean celebrity endorsement? Will a celeb endorsement or sports event on TV be counted twice? 3. p. 7 "'everyday' for television advertising equated to 28 instances of awareness over four weeks (i.e. seven days a week multiplied by four)." Why not use 4.3, the average # of weeks in a month? 4. p. 7. Type of branded merchandise: Can you provide some references for the ownership of alcohol brand merchandise question that was used? I haven't seen this wording much – specifically the identification of "drinks glasses" as a form of brand merchandise a teen could own. Research on the most common types of branded merchandise reportedly owned by youth found 88% of merchandise falls into the category of clothing items like t-shirts and jackets (64%), and headwear, such as hats and headbands (24%). "The remaining items included a wide array of alcohol-branded paraphernalia, such as jewelry, key chains, shot glasses, posters, and pens." (McClure et al. 2009). My concern is if their family cabinet contains a pint glass with a beer logo on it, will the teen consider that his own glass and affirmatively answer the question? Should that count as branded merchandise ownership? Or is that confounded with parental consumption? 5. p. 8. Is the total AUDIT-C score a summation of the three composite parts (frequency of consumption, units drunk & frequency of high episodic drinking)? If so, is it possibly double counting high episodic drinkers? For instance, if a female reported drinking 4 or more times a week for frequency of consumption, at 6
---

	units each time, and then reported daily or almost daily frequency of high episodic drinking, does she end up with a score of 4 (frequency of consumption) + 2 or 3 (whichever corresponds to 6 drink units) + 4 (daily or almost daily high episode drinking) = 10-11? 6. P. 9. Did you conduct any sensitivity analyses to compare the coefficients of interest from analysis with and without weights, or look at variance explained when weights and interactions were included in determining to use a model-based vs weighted approach for the regression analyses? 7. p. 11. How does the % in each country compare with true population size? 8. Do you think respondents consider alcohol-industry posts on social media to be adverts? On most platforms, brands have official pages which they post to (for free), and then can run targeted ads separately. 9. p. 13. I expected to see associations between branded merchandise ownership and other variables here, as in the other sections. 10. p. 13. Why was the reference group for age of initiation 14-15 for the linear regression, and 13 or under for logistic? 11. P. 16. I thought sports was a channel unique from tv – this example mentions tv & celebs, then draws a conclusion about sports....? Clarifying if the channels are mutually exclusive could help. 12. P. 16. On the lack of an association between marketing awareness with susceptibility - research (as mentioned earlier in the article) has also found that not all marketing is equal, or equally appealing to youth. Lumping all marketing together might be hiding positive associations (also, see Padon et al. 2018 Assessing Youth-Appealing Content in Alcohol Advertisements: Application of a Content Appealing to Youth (CAY) Index, Health Commun.)
--	---

VERSION 1 – AUTHOR RESPONSE

1. Reviewer One: Jonathan Noel

1.1. Paragraph 1, sentence 1 - Age ranges would be useful here because of the intention to use "young people" as a general term for both adolescents and young adults.

Our response: In the introduction we now clarify:

“Adolescents and young adults (hereafter ‘young people’, aged 11- 19 years old) are a focal population...” (Pg. 4; Line 6).

1.2. Last paragraph - A description of the U.K. regulatory environment would be useful in the Introduction to better orient the reader. This should include any literature demonstrating the effectiveness or ineffectiveness of the systems used.

Our response: In the introduction we now include a brief summary of the self-regulatory approach employed in the UK, and references which have queried its efficacy and effectiveness. Specifically we say:

“In the UK, the influence of alcohol marketing on young people has been a topic of debate for decades [9,28]. These debates are further supplemented by concerns about the efficacy and effectiveness of self-regulation, the predominant approach employed to control alcohol marketing in the UK. This includes suggestions that self-regulation provides inadequate restrictions, is not consistently enforced or complied with, is retrospective and slow to react to complaints, lacks meaningful sanctions, and lags behind modern marketing methods [9,28-32].” (Pg. 5; Lines 22-25 and Pg. 6; Lines 1-2).

1.3. Design and Sample, sentence 1 - Does the age range of 11-19 year olds correspond to the authors' definition of adolescents and young adults?

Our response: We now clarify at the beginning of the manuscript that our definition of adolescents and young adults covers those aged 11-19 years old, and this is congruent to the sample in this study. (Pg. 4; Line 6).

1.4. Demography - What was the purpose of collecting each of these variables? Why are they important to adjust for in the final analysis?

Our response: In the methods we now clarify:

“Alcohol consumption is not homogeneous among young people in the UK [4-7]. It is therefore important to adjust for demographic variation when examining any factors purported to be associated with consumption.” (Pg. 7; Lines 6-8).

1.5. Awareness of alcohol marketing, paragraph 2 - It would be useful to provide an example of how awareness was operationalized for lower exposure than the "everyday" category.

Our response: In the methods, we now clarify how a lower frequency of alcohol marketing awareness would have translated over the past month:

“For example, an answer of ‘everyday’ equated to 28 instances of awareness over four weeks (i.e. seven days per-week multiplied by four) and 1-2 times per-week equated to six instances over four weeks (i.e. 1.5 times per-week multiplied by four) (see Table 1 for other response options).” (Pg. 8; Lines 7-10).

1.6. Awareness of alcohol marketing, paragraph 2 - Several studies have aggregated values across different media to determine awareness of alcohol marketing. Did the authors' use a previous validated aggregation approach (which would require citations) or was a method created specifically for the study (which would require further explanation about why this method is reliable and valid)?

Our response: We thank the reviewer for their important comment concerning how we estimated marketing awareness. The method of aggregation was devised specifically for this study (or, at least, we have no knowledge of it being used previously). It was developed to address some of the important limitations of previous methods of estimating exposure, most notably the likelihood of

underestimating awareness. We now explain that, compared to previous approaches employed in the UK, the current method likely provides a more reliable and accurate estimate of awareness. In support, we include citations for recent research which has also used Likert scales to measure frequency of awareness of marketing, and also citations of research which has aggregated exposure over a set time period (as per this study). Specifically, in the methods, we say:

“In the UK, survey research which has measured awareness of alcohol marketing has typically used dichotomous response options for each channel (e.g. Yes/No) and used a summation across these to estimate overall awareness. [33,39]. This method, however, only provides insight into breadth of marketing awareness across channels, not frequency or volume, and therefore lacks sensitivity and may underestimate awareness. To enhance accuracy in this study, the self-reported frequency of awareness for each marketing example was converted into the estimated number of days that marketing had been seen in a four-week period (i.e. ‘one month’). This timeframe is consistent with previous research [40,41] and is representative of the minimum number of days in any month. For example, an answer of ‘everyday’ equated to 28 instances of awareness over four weeks (i.e. seven days per-week multiplied by four) and 1-2 times per-week equated to six instances over four weeks (i.e. 1.5 times per-week multiplied by four) (see Table 1 for other response options). Scores across the nine channels were summed to create an aggregate score, providing an approximation of total alcohol marketing awareness in the past month. Estimating total volume of awareness, as opposed to breadth across channels, is consistent with other recent alcohol marketing research [42,43].” (Pg. 7; Lines 24-25 and Pg. 8; Lines 1-13).

1.7. Awareness of alcohol marketing, paragraph 2, sentence 4 - What is the rationale for only including cases where there was a valid answer for all channels? The values from these participants would most likely be a conservative measure of exposure and are unlikely to bias the results towards a statistically significant finding. I suggest conducting a sensitivity analysis including these cases to determine the effects on the results.

Our response: As part of developing a more sensitive approach for estimating overall marketing awareness, we wanted to eliminate as many factors as possible which may confound accuracy. As pointed out by the reviewer, indicating ‘not sure’ to any of the channels meant that a respondent’s potential aggregated score was, by default, more conservative than those who provided a valid answer to all channels. As we considered an accurate estimate of awareness to be of paramount importance in this study, including those with ‘not sure’ awareness to any channel may have therefore biased the proportion of respondents considered to have ‘low’ or ‘medium’ awareness (or indeed what thresholds were considered low or medium) and may have confounded between-group comparisons or multivariate analyses. In response, we now elaborate on our rationale for excluding these cases in the methods:

“In this study, an aggregate awareness score was only computed when a valid answer had been given for all nine marketing examples. To provide meaningful interpretative utility, the aggregate score for the valid sample was split into tertiles of low (aggregate score <16; awareness approximately every other day), medium (17-53; awareness approximately daily), and high awareness (>54; awareness almost twice daily). If a participant answered ‘not sure’ to any of the nine channels they were coded as ‘not stated’ for the aggregate score. Indicating ‘not sure’ meant that a respondent’s potential aggregate score was, by default, more conservative than those who provided a valid answer to all nine examples. These respondents were therefore coded as a separate ‘not sure’ category to avoid biasing the proportion of valid respondents considered to have low or medium awareness, or what the tertiles boundaries were.” (Pg. 8; Lines 14-23).

While we agree that a sensitivity analysis would have enabled us to compare the influence of the separate ‘not sure’ category, compared to the aggregated awareness for all respondents, we feel that

this may have undermined the robustness of the results. Even if no difference was observed in a sensitivity analysis, the findings would still have a known confounding influence on accuracy. We agree, however, that accounting for the potential influence of 'not sure' cases (and their relative characteristics) is important to show a robust association between alcohol marketing and consumption-related outcomes, and therefore this is why the 'not sure' marketing awareness category is included in each of the multivariate models (Tables 4, 5 and 6). It is also noted that the frequency of awareness reported in table 1, and the summary of these results described in text, refers to all respondents (including 'not sure') (Table 1 and Pg. 13; Lines 15-24). Those who indicated 'not sure' are only excluded from the bivariate analyses reported in table 2.

1.8. Awareness of alcohol marketing, paragraph 2, last sentence - What was the rationale for using tertiles to categorize marketing awareness?

Our response: Categories, based on tertiles of aggregate awareness, provided meaningful interpretation for the different levels of alcohol marketing awareness in young people. For example, low marketing awareness (<17 instances per month) roughly equated to seeing alcohol marketing approximately every other day, while high awareness (>54 instances per month) roughly equated to seeing alcohol marketing almost twice every day. These thresholds were considered to provide greater sensitivity and interpretative utility compared to either binary categories or more detailed categories (for example: very low, low, medium, high and very high). Dividing the sample into categories was also essential in order to account for the 'not sure' category, for who an aggregate score could not be computed. In response, we now provide a more explicit rationale for using tertiles in the methods:

"To provide meaningful interpretative utility, the aggregate score for the valid sample was split into tertiles of low (aggregate score <16; awareness approximately every other day), medium (17-53; awareness approximately daily), and high awareness (>54; awareness almost twice daily)." (Pg. 8; Lines 14-18).

1.9. Ownership - Where did the question on ownership of alcohol brand merchandise originate from? Has it been validated previously?

Our response: In the methods, we now indicate that we adapted our measure of owning branded merchandise from those used in previous research:

"Ownership of alcohol branded merchandise was measured through a single item adapted from previous research [33,44]. Participants were asked 'Do you own any merchandise (such as clothing or drinks glasses) that show an alcoholic drink brand or logo?' (Yes/No/Not sure)." (Pg. 9; Lines 1-3).

We provide a fuller discussion of the development and design of the question, and cognitive testing, in response to a query raising by reviewer two (see response to reviewer 2.4).

1.10 Drinking status - Where did the drinking status questions originate from? Have they been validated previously?

Our response: The item used to measure drinking status was lifted from a previous longitudinal study of alcohol marketing and youth consumption in the UK (Gordon et al., 2011; Gordon et al., 2010). The measure was originally adapted from a long-running, repeat cross-sectional, survey of youth experiences of tobacco marketing and smoking in the UK, the Youth Tobacco Policy Survey (e.g. Bauld et al., 2017). As part of its deployment in these two studies, the measure was cognitively tested with young people. Further cognitive testing was also undertaken with 100 young people before the

current survey was used in the field to ensure age and cultural comprehension. In the text, we now provide citations to support where the question has been used previously (Pg. 9; Lines 5-8).

References not cited in the manuscript:

Bauld, L., MacKintosh, A.M., Eastwood et al. (2017). Young people's use of e-cigarettes across the United Kingdom: Findings from five surveys 2015-2017. *International Journal of Environmental Research and Public Health*. 14(9), e973.

1.11. Alcohol consumption - Was the AUDIT-C given to all participants or only those that were considered lifetime drinkers?

Our response: All respondents completed the first item on the AUDIT-C, regardless of whether they were classed as 'ever' or 'never' drinkers based on the drinking status question. However, those who answered 'never' for frequency of consumption on the first AUDIT-C item were classed as 'non-drinkers' and routed to the next question. All those who answered anything other than 'never' were classed as 'current drinkers' and were asked to complete the final two AUDIT-C items. We now realise that including assessment of current drinking status under the same heading of ever drinking (as we did in the original manuscript) is misleading and unnecessarily repetitive. In the methods we now clarify:

"Those who answered anything other than 'never' on the first AUDIT-C item were classed as current drinkers and asked to complete the final two items. All other respondents (i.e. those stating 'never' for frequency of consumption) were classified as non-drinkers and were not asked to complete the final two items. In current drinkers, a total AUDIT-C score was computed by summing the three AUDIT-C items (0-12), with a cut-off of >5 used to identify higher-risk consumption [45]." (Pg. 9; Lines 19-24).

1.12. Confounding variables - I agree that these are potential confounders, but it is important to include citations that conclude these are important factors in alcohol consumption.

Our response: In the methods we now include a rationale and citations which support the use of these covariates:

"Confounding factors, reported to influence consumption in young people and used in previous alcohol marketing research, were included as covariates to contextualise any association between marketing and consumption [33,34,46,47]." (Pg. 10; Lines 9-11).

1.13. Confounding variables, sentence 3 - What is the justification or who promoted the idea of converting self-reported consumption into number of days alcohol was consumed each year? That appears to have the potential to inflate alcohol consumption.

Our response: We thank the reviewer for raising this point. Having revisited our analysis, we realise that the data for frequency of parental and peer drinking was analysed in relation to either the nine scale categories as per the original survey (in the Chi-square Bivariate analyses) or, in the multivariate regressions, by five collapsed categories. We had intended to use the estimated number of days on which alcohol was consumed as continuous variables in the linear regression, but realised that categorisation was needed to facilitate comparison to those who indicated 'not stated'. This statement about conversion of parental and peer consumption into days is therefore redundant and has been replaced with a description of the collapsed categories:

"Frequency of consumption was measured for the mother (female carer), father (male carer), and closest friend (each scored: 1=Never – 9=Every day or almost every day; Prefer not to say; Not

applicable). For all three groups, consumption was collapsed into five categories (Never, Less than monthly, Monthly or Fortnightly, At Least weekly, and Not Stated).” (Pg. 10; Lines 11-15).

1.14. Analysis, sentence 1 - Please include the corporate information for SPSS.

Our response: In the analysis we now say “Data were analysed using SPSS version 23 (SPSS Inc., Chicago IL).” (Pg. 11; Lines 10).

1.15. Analysis, sentence 2 - If this was a representative sample, what is the purpose of weighting the data? What characteristics factored into the weights? Was a sensitivity analysis conducted with and without the weights to determine the impact?

Our response: We thank the reviewer for raising this point and we agree that greater clarity was required on the survey weights. Although the sample was recruited to be broadly representative of the UK population, the ability to recruit a fully representative sample was constrained by the composition and response from YouGov’s panel members. For example, some demographic groups may have been under-represented in the panel in general, while others may have ended up being under-represented if not able to respond within the timeframe. In order to correct for any potential under-representation, YouGov provide weights for each participant to enable the data to be aligned with the characteristics of the UK population. This approach has been widely cited in previous research using YouGov panel data (Hooper et al. 2017; Moodie et al., 2018; Rosenberg et al., 2017). In the methods, we now clarify:

“The survey was hosted by YouGov, a market research company, who recruited a sample intended to be representative of the UK population from their UK panel [35]. Participants aged 16 or over were approached directly to participate, while those aged under 16 were approached through existing adult panel members known to have children. A survey weight was provided for each respondent (based on age, gender, ethnicity, region, and social grade) to enable descriptive data to be representative of the UK population.” (Pg. 6; Lines 17-24).

Our response to conducting analysis with, and without, survey weights is addressed in relation to the next comment.

References not cited in manuscript:

Hooper, L., Anderson, A.S., Birch, J., Forster, A.S., Rosenberg, G., Bauld, L., & Vohra, J. (2017). Public awareness and healthcare professional advice for obesity as a risk factor for cancer in the UK: A cross-sectional survey. *Journal of Public Health*. [Advance online publication]. Available at: <https://academic.oup.com/jpubhealth/advance-article/doi/10.1093/pubmed/fox145/4582914?searchresult=1>.

Moodie, C., MacKintosh, A.M., Thrasher, J.F., McNeill, A., & Hitchman, S. (2018). Use of cigarettes with flavour-changing capsules among smokers in the United Kingdom: An online survey. *Nicotine and Tobacco Research*. [Advance online publication]. Available at: <https://academic.oup.com/ntr/advance-article/doi/10.1093/ntr/nty173/5081538>.

Rosenberg, G., Bauld, L., Hooper, L., Buyk, P., Holmes, J., Vohra, J. (2017). New national alcohol guidelines in the UK: Public awareness, understanding and behavioural intentions. *Journal of Public Health*. 40(3), 549-556.

1.16. Analysis, paragraph 2, sentence 1 - This is not an appropriate rationale for conducting an analysis on unweighted data. The weighting procedure prior to analysis and the adjustment of covariates within a statistical analysis are inherently different computations that may or may not be

related to each other. I suggest conducting the analysis using the weighted data for consistency purposes.

Our response: We thank the reviewer for their comment. We acknowledge that while there is a general consensus on the use of survey weights in descriptive data (as done in the study), there is less consensus for multivariate analyses (e.g. Young & Johnson, n.d). As all the factors involved in creating the survey weights are included as covariates in the multivariate models we did not consider it appropriate or necessary to also conduct the multivariate analyses on weighted data. Our approach of conducting descriptive analysis on weighted data and multivariate analyses on unweighted data (providing factors are accounted for in the models) is consistent with previous research examining the effect of marketing in the UK, including studies based on YouGov panel data (Harris et al., 2006; Moodie et al., 2008; Moodie et al., 2016a; Moodie et al., 2018). Nevertheless, we did follow the reviewer's suggestion to run the multivariate models with and without the survey weights and found no major differences on the standardised co-efficients (linear model) and adjusted odds ratios (logistic models), particularly on the key variables of interest (e.g. marketing awareness). In the methods we now clarify:

"All multivariate analyses were conducted on unweighted data as the factors used to construct the weights were included as covariates in the multivariate models. The multivariate analyses were repeated on weighted data to check for consistency. As results for the key independent variables (marketing awareness and ownership of branded merchandise) were consistent, only the unweighted results are presented." (Pg. 12; Lines 21-25).

References not cited in manuscript:

Harris, F., MacKintosh, A.M., Anderson, S., Hastings, G., Borland, R., Fong, G.T., Hammond, D., & Cummings, K.M. (2006). Effects of the 2003 advertising/promotion ban in the United Kingdom on awareness of tobacco marketing: Findings from the International Tobacco Control (ITC) Four Country Survey. *Tobacco Control*, 15, 26-33.

Moodie, C., MacKintosh, A.M., Brown, A., & Hastings, G. (2008). Tobacco marketing awareness on youth smoking susceptibility before and after an advertising ban. *European Journal of Public Health*, 18(5), 484-490.

Moodie, C., MacKintosh, A.M., Gallopel-Morvan, K., Hastings, G., & Ford, A. (2016a). Adolescents' perceptions of an on-cigarette warning. *Nicotine and Tobacco Research*, 19(10), 1231-1237.

Moodie, C., Sinclair, L., MacKintosh, A.M., Power, E., & Bauld, L. (2016). How tobacco companies are perceived within the United Kingdom: An online panel. *Nicotine and Tobacco Research*, 18(1), 1766-1772.

Moodie, C., MacKintosh, A.M., Thrasher, J.F., McNeill, A., & Hitchman, S. (2018). Use of cigarettes with flavour-changing capsules among smokers in the United Kingdom: An online survey. *Nicotine and Tobacco Research*. [Advance online publication].

Young, R. & Johnson, D.R. (n.d). To weight or not to wright, that is the question: Survey weights and multivariate analysis. Pennsylvania, PA: Pennsylvania State University Population Research Institute. Available at: http://www.aapor.org/AAPOR_Main/media/AnnualMeetingProceedings/2012/03_-Young-Johnson_A2_Weighting-paper_aapor-2012-ry.pdf.

1.17. Analysis, paragraph 2, sentence 2 - It may be better to state "multivariable linear regression." Upon reading the term "hierarchical," my first reaction was to think about multi-level modeling. It was a

quick over thinking on my part but just double check these terms to ensure accuracy and comprehension.

Our response: We agree with the reviewer that use of the term hierarchical is potentially misleading. Throughout the manuscript we now refer to 'multivariate linear regression' or 'multivariate logistic regressions' as opposed to hierarchical.

1.18. Analysis, paragraph 3, sentence 4 - State the reference categories in the methods.

Our response: We thank the reviewer for this comment. While we agree that it is best practice to clearly outline all the control variables (covariates) when describing multivariate analysis, we do not feel it is necessary to report each of the reference categories. Due to wider changes requested in the review process, and because we would need to report the reference categories for around 15 variables, we do not feel it would not be an efficient use of space to duplicate information already reported in the tables in the text. We do, however, indicate that "Reference categories for each variable are displayed in results." (Pg. 12; Lines 4-5; Pg. 12; Lines 10-11).

1.19. Owning alcohol branded merchandise - After all the detail of the previous section, this part seems to be lacking. Were there any differences across demographic categories?

Our response: We thank the reviewer for raising this point. We now repeat the bivariate analyses conducted for alcohol marketing awareness for ownership of alcohol branded merchandise and provide a new table summarising this information:

"Almost a fifth of participants (17%) reported owning alcohol branded merchandise. Bivariate Chi-square tests found that ownership of branded merchandise was significantly associated with being of white British ethnicity, of legal purchase age, a current drinker, a higher-risk drinker, not in education, in employment, and perceiving that parents and peers would consider it okay to consume (Table 3). Ownership of branded merchandise was also associated with greater frequency of mother (female carer) consumption, $\chi^2(8)=44.11$, $p<0.001$, greater frequency of father (male carer) consumption, $\chi^2(8)=56.49$, $p<0.001$, and greater frequency of close friends drinking, $\chi^2(8)=178.76$, $p<0.001$. There was also an overall effect of IMD, $\chi^2(4)=15.73$, $p<0.01$, although this had no distinct pattern across escalating deprivation. There was no difference by resident country or gender." (Pg. 15; Lines 3-13; Table 3).

1.20. Throughout the Methods, the results of alcohol marketing awareness and ownership of branded merchandise should be positioned at the top of each section, not at the bottom. Those variables are the reason for conducting the study. highlight them.

Our response: We thank the reviewer for this comment. We assume the start of their comment is meant to refer to 'throughout the models' as opposed to 'through the methods'? Assuming we have interpreted this correctly, we now include discussion of any association with alcohol marketing at the beginning of each section describing the multivariate analysis (Pg. 15; Lines 17-25; Pg. 16; Lines 9-20; Pg. 17; Lines 1-10).

1.21. Refrain from repeating the bulk of the information from the tables. Be concise and only highlight information stemming from the study hypotheses or novel findings.

Our response: We agree with the reviewer that streamlining reporting will enhance readability. For each multivariate model, we now only report the significance for the demographic and confounding variables, which are secondary to the aim of the study. For these variables we no longer report the coefficients (and 95% CIs) for the linear regression in-text and no longer report the adjusted odds

ratio (and 95% CIs) for the two logistic regressions in-text, as this information is duplicated in the tables. We do still report this information for alcohol marketing variables, as these are the primary variables of interest.

We have also edited the description of the models throughout to reduce the bulk of text to increase clarity and readability (Pg. 15; Lines 17-25; Pg. 16; Lines 9-20; Pg. 17; Lines 1-10).

1.22. Re-think phrasing and sentence structure in a few locations. Several sentences start with "That" and I often had to re-read a sentence to fully comprehend its meaning.

Our response: We thank the reviewer for this comment and acknowledge that some of the syntax and phrasing in the discussion complicated readability. As part of wider edits of the manuscript we have sought to identify and address these. For example, rather than one long sentence we have revised the opening of the discussion to read:

"The findings indicate that young people in the UK are aware of a variety of alcohol marketing and almost a fifth own branded merchandise. The results also show that awareness of marketing and ownership of branded merchandise is associated with increased consumption and higher-risk drinking in current drinkers, and that ownership of branded merchandise is associated with susceptibility in never-drinkers." (Pg. 17; Lines 15-22).

1.23. Paragraph 5 - It would be useful to better differentiate between the limitations of the current study and future research studies that could be conducted. I understand the relationship but they are different entities.

Our response: We agree with the reviewer that, while the suggestions for future research are interesting, presenting these alongside the study limitations was complicated. As they not integral to the discussion, and due to wider edits to the manuscript to maintain an appropriate word length, the suggestions for future research have been removed.

1.24. I suggest placing the results for marketing awareness and alcohol branded merchandise at the top of the tables.

Our response: We thank the reviewer for this suggestion. While we agree that it makes sense to report any association of the alcohol marketing variables earlier in the written descriptions of the multivariate analyses (see response to comment 1.20) we do not feel that the marketing variables should be included at the top of the multivariate tables. Our rationale is two-fold. First, by positioning them at the bottom of the table, this clarifies that any association between marketing and the consumption exists after controlling for the above covariates. Second, reporting the marketing variables at the bottom of the table is congruent to other consumer research exploring the impact of marketing on both tobacco and alcohol consumption in young people, this includes alcohol marketing research in the UK (Gordon et al., 2011) and also studies reported in the current journal (Ford et al., 2013). We therefore feel it is appropriate to continue this convention for readability, and hope the reviewer will understand.

2. Reviewer Two: Alisa Padon

2.1. Was there any eligibility criteria? What was the overall response rate? Was parental permission required?

Our response: We thank the reviewer for their comment. The only eligibility criteria was that the respondents either needed to be part of YouGov's UK panel or the child of an adult who was part of

the panel. All children under the age of 16 years old were recruited through their parents. All those aged 16 or older were contacted directly.

In the methods, we now clarify: “Data come from the 2017 Youth Alcohol Policy Survey, an online cross-sectional survey conducted with 11-19-year olds in the UK (n=3,399). Responses were collected April–May 2017. The survey was hosted by YouGov, a market research company, who recruited a sample intended to be representative of the UK population from their UK panel [35]. Participants aged 16 or over were approached directly to participate, while those aged under 16 were approached through existing adult panel members known to have children. A survey weight was provided for each respondent (based on age, gender, ethnicity, region, and social grade) to enable descriptive data to be representative of the UK population. The study design was informed by previous cross-sectional surveys in the UK which have explored young people’s experiences of alcohol and tobacco marketing [33,36].” (Pg. 6; Lines 17-24).

2.2. Are the channels mutually exclusive? Does this “channel”: “(6) famous people in films, music videos or TV or pictured in magazines;” mean celebrity endorsement? Will a celeb endorsement or sports event on TV be counted twice?

Our response: We thank the reviewer for raising this point. The nine examples of marketing measured in this study are intended to be mutually exclusive. For each example, even when marketing could be operationalised in different ways (e.g. sponsorship could be of either a football team or music event) respondents provided only one answer to indicate their overall frequency of awareness. For the celebrity endorsement question, for example, only one response was given for any recall of ‘famous people in films, music videos, on TV or pictured in magazines with alcohol’ and not for each of the constituent parts. In the process of creating the aggregated score, all examples of marketing were treated equally (i.e. none were counted twice). In response, we have amended the wording to better reflect the mode of response:

“Awareness of alcohol marketing was assessed through structured, self-reported recall, a method frequently used in consumer research [33]. Participants were prompted with the statement ‘Over the last month, how often, if at all, have you seen...’ and then presented with descriptions of nine examples of alcohol marketing: (1) newspapers or magazines; (2) television; (3) billboards; (4) radio; (5) adverts on social media (e.g. YouTube, Tumblr, Facebook, SnapChat, Instagram or other social media); (6) famous people in films, music videos or TV or pictured in magazines with alcohol [celebrity endorsement]; (7) sports, games, or events sponsorship; (8) special price offers; and (9) competitions or prize draws. As per recent research [37,38], a Likert scale was used to measure frequency of noticing marketing in the past month for each of the nine examples (1=Everyday – 6=Not in the past month; Not sure).” (Pg. 7; Lines 14-23).

2.3. p. 7 “‘everyday’ for television advertising equated to 28 instances of awareness over four weeks (i.e. seven days a week multiplied by four).” Why not use 4.3, the average # of weeks in a month?

Our response: We thank the reviewer for raising this point. Our rationale for using a four-week period (i.e. 28 days) was two-fold. First, this timeframe is consistent with some previous research into alcohol marketing exposure in young people (Siegel et al. 2015; Synder et al. 2006). Second, while other studies have used longer intervals for measuring exposure (e.g. 30 days; Ross et al., 2015), 28 represents the minimum number of days in any given month. While we understand the potential enhanced accuracy that could be gained by using 4.3 weeks, we do not feel this is congruent to the wider literature, and that the current approach – albeit potentially marginally more conservative – is easily interpreted by the reader. In response, we have added a sentence to the methods to clarify our rationale for choosing 28 days, including the above citations:

“To enhance accuracy in this study, the self-reported frequency of awareness for each marketing example was converted into the estimated number of days that marketing had been seen in a four-week period (i.e. ‘one month’). This timeframe is consistent with previous research [40,41] and is representative of the minimum number of days in any month. For example, an answer of ‘everyday’ equated to 28 instances of awareness over four weeks (i.e. seven days per-week multiplied by four) and 1-2 times per-week equated to six instances over four weeks (i.e. 1.5 times per-week multiplied by four) (see Table 1 for other response options).” (Pg. 8; Lines 3-10).

References not cited in manuscript:

Ross, C., Maple, E., Siegel, M., DeJong, W., Naimi, T.S., Ostroff, J., Padon, A.A., Borzekowski, D.L.G., & Jernigan, D.H. (2016). The relationship between brand-specific alcohol advertising on television and brand-specific consumption among underage youth. *Alcohol: Clinical and Experimental Research*, 38(8), 2234-2242.

2.4. p. 7. Type of branded merchandise: Can you provide some references for the ownership of alcohol brand merchandise question that was used? I haven’t seen this wording much – specifically the identification of “drinks glasses” as a form of brand merchandise a teen could own. Research on the most common types of branded merchandise reportedly owned by youth found 88% of merchandise falls into the category of clothing items like t-shirts and jackets (64%), and headwear, such as hats and headbands (24%). “The remaining items included a wide array of alcohol-branded paraphernalia, such as jewelry, key chains, shot glasses, posters, and pens.” (McClure et al. 2009). My concern is if their family cabinet contains a pint glass with a beer logo on it, will the teen consider that his own glass and affirmatively answer the question? Should that count as branded merchandise ownership? Or is that confounded with parental consumption?

Our response: We thank the reviewer for raising the important query of how this question may have been interpreted by participants. Previous research has measured ownership of branded merchandise in a variety of ways, including single item questions and those which measure ownership of several forms of merchandise (e.g. clothing, hats, and lighters).

In our study, the question and wording was adapted from previous research (Gordon et al. 2011; McClure et al. 2013), including a review of survey research on branded merchandise (Jones et al. 2016). As space was limited, it was not possible to examine ownership of branded merchandise to the same degree of detail as awareness of marketing, and a single item measure was required. While we intended the question to be deliberately broad, to enable young people to consider the variety of forms that branded merchandise may take, we also felt it necessary to provide some provide examples to prompt recall and aid comprehension. In the UK, alcohol glassware is reported as a salient and effective marketing technique (Stead et al., 2014) and, as the reviewer alludes, it has been cited in previous research (McClure et al., 2009). Drinks glasses were therefore considered an appropriate example. The survey was cognitively tested with 100 young people to check the cultural and age appropriateness of the measures, and the inclusion of drinks glasses as an example of branded merchandise was not flagged as confusing or inappropriate. The reviewer’s comment about the stage at which young person would consider branded merchandise to be something ‘they’ own (i.e. items found in the family home) is an interesting and important point, but beyond the scope of this current research.

The two forms of merchandise incorporated into the question were purely intended to be independent examples. Nevertheless, we agree with the reviewer’s comment that providing any example, or not explicitly asking young people to consider other forms of merchandise, may have influenced how they respond to this item. In response, we have made two changes:

In the methods, we now include citations of the research that was used to develop this question: “Ownership of alcohol branded merchandise was measured through a single item adapted from previous research [33,44]. Participants were asked ‘Do you own any merchandise (such as clothing or drinks glasses) that show an alcoholic drink brand or logo?’(Yes/No/Not sure). (Pg. 9; Lines 1-3).

In the discussion of limitations, we now also note that providing any examples may potentially bias recall: “Finally, measurement of owning branded merchandise also only included two examples as prompts (clothing and drinks glasses). It is possible that different prompts may have altered recall, and that multiple items or a free text response option would have provided greater clarity on merchandise owned. (Pg. 20; Lines 1-7).

References not cited in manuscript:

McClure, A.C., Stoolmiller, M., Tanski, S.E., & Worth, K.A. (2009). Alcohol-branded merchandise and its association with drinking attitudes and outcomes in US adolescents. *Archives of Pediatric and Adolescent Medicine*, 163, 211-217.

Stead, M., Angus, K., MacDonald, L., & Bauld, L. (2014). Looking into the glass: Glassware as an alcohol marketing tool and the implications for policy. *Alcohol and Alcoholism*, 49(3), 317-320.

2.5. p. 8. Is the total AUDIT-C score a summation of the three composite parts (frequency of consumption, units drunk & frequency of high episodic drinking)? If so, is it possibly double counting high episodic drinkers? For instance, if a female reported drinking 4 or more times a week for frequency of consumption, at 6 units each time, and then reported daily or almost daily frequency of high episodic drinking, does she end up with a score of 4 (frequency of consumption) + 2 or 3 (whichever corresponds to 6 drink units) + 4 (daily or almost daily high episode drinking) = 10-11?

Our response: The AUDIT-C is a summation of the three composite parts. In the methods, we now clarify “In current drinkers, a total AUDIT-C score was computed by summing the three AUDIT-C items (0-12), with a cut-off of >5 used to identify higher-risk consumption [45].” (Pg. 9; Lines 22-24).

To clarify the reviewer’s example, a female participant drinking four or more times per week, consuming 5-6 units in a typical session, and reporting daily or almost daily heavy episodic drinking would score 10 on the AUDIT-C (4+2+4, respectively). The intention of the AUDIT-C is provide a total score reflective three types of consumption behaviour (frequency, quantity, and heavy episodic drinking) and it also can be used to classify those consuming at higher and lower risk. Although we understand, and appreciate, the reviewer’s comment that the scale provides two means of calculating frequency of heavy episodic drinking (a summation of the first two items or the third item in isolation), the AUDIT-C has been previously validated in the UK (Khadjesari et al. 2018) and in adolescents (Liskola et al. 2018; Rumpf et al. 2013). Furthermore, as heavy episodic drinking was not the outcome in the multivariate analyses (only a constituent part of the total score of classification of higher-risk drinking), we do not feel that the potential to estimate this outcome in two ways to be problematic.

References not cited in manuscript:

Khadjesari, Z., White, I.R., McCambridge, J., Marston, L., Wallace, P., Godfrey, C., & Murray, E. (2017). Validation of the AUDIT-C in adults seeking help with their drinking online. *Addiction Science and Clinical Practice*, 12(2).

Liskola, J., Haravuori, H., Lindberg, N., Niemela, S., Karlsson, L., Kiviruusu, O., & Marttunen, M. (2018). AUDIT and AUDIT-C as screening instructions for alcohol problem use in adolescents. *Drug and Alcohol Dependence*, 188, 266-273.

Rumpf, H.J., Wohler, T., Freyer-Adam, J., Grotheus, J., & Bischof, G. (2013). Screening questionnaires for problem drinking in adolescents: Performance of AUDIT, AUDIT-C, CRAFFT, and POSIT. *European Addiction Research*, 19, 11-127.

2.6. Did you conduct any sensitivity analyses to compare the coefficients of interest from analysis with and without weights, or look at variance explained when weights and interactions were included in determining to use a model-based vs weighted approach for the regression analyses?

Our response: Please refer to our response to reviewer comment 1.16 for discussion of analysis with and without survey weights. In brief, we did conduct the multivariate analyses on both weighted and unweighted data and found no major differences in the standardised co-efficients (linear regression) and adjusted odds ratios (logistic regressions), particularly for the key variables of interest. This is now reported in the methods (Pg. 12; Lines 21-25).

2.7. How does the % in each country compare with true population size?

Our response: YouGov recruited a sample intended to be representative of the UK population. In this study the proportion of young people in each country is reflective of the distribution of the true population size across the UK. For example, in our sample, 84% were from England, 8% Scotland, 5% Wales, and 3% Northern Ireland. This is consistent with the latest estimates on true population size. The UK population is estimated at 66 million. Within this 55.6 million are in England (84%), 8% are in Scotland (5.4 million), 5% are from Wales (3.1 million), and 3% are in Northern Ireland (1.9 million) (Office for National Statistics, 2018).

References not cited in the manuscript:

Office for National Statistics. (2018). Overview of the UK population: November 2018.: <https://www.ons.gov.uk/peoplepopulationandcommunity/populationandmigration/populationestimates/articles/overviewoftheukpopulation/november2018>.

2.8. Do you think respondents consider alcohol-industry posts on social media to be adverts? On most platforms, brands have official pages which they post to (for free), and then can run targeted ads separately

Our response: We agree that alcohol companies can market their products on social media in a variety of sophisticated and inter-linked ways. This ranges from paid-for-media (e.g. paid promoted adverts), owned media (e.g. posts on their own social media pages), and earned media (e.g. user-generated branding such as 'fan posts' or comments) (Critchlow et al., 2016). Furthermore, as the design of social media is not homogeneous, the opportunities for marketing vary across each platform.

In this study, 'adverts for alcohol on YouTube, Tumblr, Facebook, Snapchat, Instagram or other social media' was used as a universal term to capture awareness of all commercially-generated content on these digital platforms. This universal term was required due to limitations on available space in the survey, which meant that it was not possible to include measures that reflect awareness of all the varied forms that marketing can take (e.g. posts on a brand operated page vs. paid for video adverts). We note in the limitations that we only provide limited understanding of digital marketing platforms (Pg. 19; Line 21-22).

Although there is existing research on what young people recognise to constitute alcohol marketing on social media (Atkinson et al., 2016; Lyons et al., 2014; Purves et al., 2018), understanding what forms of content young people recalled when self-reporting their awareness is beyond the scope of

this study. We do note, however, that another publication from project will specifically focus on participation with alcohol marketing on social media (Critchlow et al., under review) and that survey was cognitively tested with 100 young people to cultural and age appropriateness to respond.

References not cited in the manuscript:

Critchlow, N., MacKintosh, A.M., Hooper, L., Thomas, C., Vohra, J. (under review, provisionally accepted). Participation with alcohol marketing and user-created promotion on social media and the association with higher-risk alcohol consumption and brand identification among adolescents in the UK. *Addiction Research and Theory*.

Lyons, A. C., McCreanor, T., Hutton, F., Goodwin, I., Barnes, H. M., Griffin, C.... Samu, L. (2014). *Flaunting it on Facebook: Young adults, drinking cultures and the cult of celebrity*. Wellington: Massey University School of Psychology.

2.9. I expected to see associations between branded merchandise ownership and other variables here, as in the other sections.

Our response: As per our response to comment 1.19, we now include bivariate analyses which explore how ownership of branded merchandise differs by demographic category or by level of drinking status. This includes a new table reporting these differences (Pg. 15; Lines 3-13; Table 3).

2.10. Why was the reference group for age of initiation 14-15 for the linear regression, and 13 or under for logistic?

Our response: We thank the reviewer for raising this point. The difference is a consequence of how the covariates were handled in the two different types of regression (Table 4 is a linear regression and Tables 5 and 6 are logistic regressions). In the linear model, we needed to select the reference category against which to create the dummy variables. We chose the middle category so that the dummy comparisons related to whether initiation occurred in early adolescence or later adolescence (relative to middle adolescence). In the logistic regression, however, we used the contrast=difference function to compare each increasing category relative to the combined previous categories. As such, the first comparison represents aged 14-15 years old vs. 13 years old or younger but the second comparison represents aged 16 years or older vs. younger (i.e. the 14-15 years old and 13 years or less categories combined). This contrast=difference function is used across all the covariates in the logistic models which have >3 categories, and thus it made sense to be consistent for age of first drink. We discuss the different methods of entering categories in the methods (Pg. 11; Lines 15-25; and Pg. 12; Lines 1-20).

2.11. I thought sports was a channel unique from tv – this example mentions tv & celebs, then draws a conclusion about sports....? Clarifying if the channels are mutually exclusive could help.

Our response: Please see our response to comment 2.2 for clarification on how the marketing channels were measured. We agree that the sentence highlighted by the reviewer in the discussion was confusing and it has been removed.

2.12. On the lack of an association between marketing awareness with susceptibility - research (as mentioned earlier in the article) has also found that not all marketing is equal, or equally appealing to youth. Lumping all marketing together might be hiding positive associations (also, see Padon et al. 2018 Assessing Youth-Appealing Content in Alcohol Advertisements: Application of a Content Appealing to Youth (CAY) Index, *Health Commun.*)

Our response: We thank the reviewer for raising the important point that not all alcohol marketing channels and brands are equally appealing to youth, and this heterogeneity is not considered when evaluating overall/aggregated alcohol marketing exposure. In response, in the discussion we now clarify:

“Nevertheless, as research suggests that not all alcohol marketing or brands are equally appealing to youth [25,50], it is possible that focusing on aggregated alcohol marketing awareness (the approach in this study) may have disguised associations between individual examples of marketing and susceptibility in never drinkers.” (Pg. 18; Lines 17-21).

VERSION 2 – REVIEW

REVIEWER	Jonathan Noel Johnson & Wales University, USA
REVIEW RETURNED	19-Nov-2018
GENERAL COMMENTS	The authors have done an excellent job of responding to the reviewer's concerns. I suggest only 1 small change. In the Results - Awareness of alcohol marketing, there are several instances of medians being described within parentheses. I suggest adding equal signs to these instances (e.g. "Median = 6" instead of "Median 6").

VERSION 2 – AUTHOR RESPONSE

Response to reviewer comments for: 'Awareness of alcohol marketing, ownership of alcohol branded merchandise, and the association with alcohol consumption, higher-risk drinking, and drinking susceptibility in adolescents and young adults: A cross-sectional survey in the United Kingdom.'

Thank you to BMJ Open, and to our reviewers, for taking the time to (re)consider our manuscript.

Please find below details of how we have responded to each comment and reference to where changes are located within the manuscript. All page numbers refer to the position as shown in the PDF proof of the resubmission.

1. Reviewer One: Jonathan Noel

1.1. In the Results - Awareness of alcohol marketing, there are several instances of medians being described within parentheses. I suggest adding equal signs to these instances (e.g. "Median = 6" instead of "Median 6").

Our response: In the results, we now include equals signs where relevant:

“The most frequent sources of marketing awareness in the past month were adverts on television (Median=6 instances per month, Inter quartile range=14), celebrity endorsement (Median=6, IQR=14), and special offers (Median=6, IQR=14) (Table 1). More than a third of respondents (range: 39-43%) had noticed marketing through these channels at least weekly. Billboard adverts (Median=2 instances per month, IQR=6), sponsorship (Median=2, IQR=6), and social media adverts (Median=2, IQR=6) were noticed less than once a week, with at least a quarter of participants (range: 27-31%) having noticed these at least weekly. Lowest awareness was for adverts in the print press (Median=0 instances per month, IQR=6), on radio (Median=0, IQR=0), and competitions (Median=0, IQR=2). For each marketing example, a fifth or more (range: 19-29%) were not sure how often, if at all, they had come across alcohol marketing. Overall, 82% had noticed marketing through at least one channel.” (Pg. 13; Lines 20-25 and Pg. 14; Lines 1-6).

2. Editor comments

2.1. Please improve the reporting of the statistics throughout your abstract. For example, please include the Chi-square values in addition to P values in the abstract.

Our response: The Chi Square p values reported in the original abstract each related to two separate tests (i.e. one for awareness of marketing and a separate one for ownership of branded merchandise, albeit both tests had the same significance). We considered that it would be confusing to report both χ^2 values if it could not be determined to which test each related. Providing the full breakdown of all tests for both awareness and ownership individually meant the abstract would have exceeded the permitted word count.

In response, we now include a statement that indicates that both marketing awareness and ownership of branded merchandise varied by drinking variables and, as per the multivariate regression, report examples of this for alcohol marketing awareness, complete with the Chi Square χ^2 values:

“Chi-square tests found that awareness of marketing and ownership of branded merchandise varied within drinking variables. For example, higher awareness of alcohol marketing was associated with being a current drinker ($\chi^2=114.04$, $p<0.001$), higher-risk drinking ($\chi^2=85.84$, $p<0.001$), and perceived parental ($\chi^2=63.06$, $p<0.001$) and peer approval of consumption ($\chi^2=73.08$, $p<0.001$).” (Pg. 2; Lines 17-21).